# Learning Predictions for Algorithms with Predictions

**Mikhail Khodak**
Carnegie Mellon University
khodak@cmu.edu

**Maria-Florina Balcan**
Carnegie Mellon University
ninamf@cs.cmu.edu

**Ameet Talwalkar**
Carnegie Mellon University
talwalkar@cmu.edu

**Sergei Vassilvitskii**
Google Research - New York
sergeiv@google.com

## Abstract

A burgeoning paradigm in algorithm design is the field of *algorithms with predictions*, in which algorithms can take advantage of a possibly-imperfect prediction of some aspect of the problem. While much work has focused on using predictions to improve competitive ratios, running times, or other performance measures, less effort has been devoted to the question of how to obtain the predictions themselves, especially in the critical online setting. We introduce a general design approach for algorithms that learn predictors: (1) identify a functional dependence of the performance measure on the prediction quality and (2) apply techniques from online learning to learn predictors, tune robustness-consistency trade-offs, and bound the sample complexity. We demonstrate the effectiveness of our approach by applying it to bipartite matching, ski-rental, page migration, and job scheduling. In several settings we improve upon multiple existing results while utilizing a much simpler analysis, while in the others we provide the first learning-theoretic guarantees.

## 1 Introduction

Algorithms with predictions, a subfield of beyond-worst-case analysis of algorithms [36], aims to design methods that make use of machine-learned predictions in order to reduce runtime, error, or some other performance cost. Mathematically, for some prediction $\mathbf{x}$, algorithms in this field are designed such that their cost $C_t(\mathbf{x})$ on an instance $t$ is upper-bounded by some measure $U_t(\mathbf{x})$ of the quality of the prediction on that instance. The canonical example here is that the cost of binary search on a sorted array of size $n$ can be improved from $\mathcal{O}(\log n)$ to $\leq U_t(\mathbf{x}) = 2 \log \eta_t(\mathbf{x})$, where $\eta_t(\mathbf{x})$ is the distance between the true location of a query $t$ in the array and the location predicted by the predictor $\mathbf{x}$ [36]. In recent years, algorithms whose cost depends on the quality of possibly imperfect predictions have been developed for numerous important problems, including caching [41, 24, 34], scheduling [30, 43], ski-rental [29, 1, 15], bipartite matching [16], page migration [22], and many more [10, 17, 36].

While there has been a significant effort to develop algorithms that use earned predictions, until very recently [13, 33] there has been less focus on actually *learning* to predict. For example, of the works listed only two on ski-rental [1, 15] and one other [16] show sample complexity guarantees, and none consider the important *online learning* setting, in which problem instances may not come from a fixed distribution. This is in contrast to the related area of data-driven algorithm design [19, 3], which has established techniques such as dispersion [6] and others [8, 11] for deriving learning-theoretic guarantees, leading to end-to-end results encompassing both learning and computation. It is also despite the fact that, as we see in this work, learning even simple predictors is in many cases a non-trivial problem.

We bridge this gap and provide a framework for obtaining learning-theoretic guarantees for algorithms with predictions. In addition to improving sample complexity bounds, we show how to learn the parameters of interest in an setting with low overall regret. We accomplish this using a two-

36th Conference on Neural Information Processing Systems (NeurIPS 2022).

Table 1: Settings we apply our framework to, new learning algorithms we derive, and their regret.

| Problem | Algorithm with prediction | Feedback | Upper bound (losses) | Learning algo. | Regret |
|---|---|---|---|---|---|
| Min. weight bipartite matching (3)*,† | Hungarian method initialized by dual $\hat{\mathbf{x}} \in \mathbb{R}^n$ | Opt. dual $\mathbf{x}^*(\mathbf{c})$ | $\mathcal{O}\left(\|\hat{\mathbf{x}} - \mathbf{x}^*(\mathbf{c})\|_1\right)$ | Proj. online gradient | $\mathcal{O}\left(n\sqrt{T}\right)$ |
| Online page migration (4)* | Lazy offline optimal for predictions $\{\hat{s}_{[j]} \sim \mathbf{P}_{[j]}\}_{j=1}^n$ | Requests $\{s_{[j]}\}_{j=1}^n$ | $\tilde{\mathcal{O}}\left(\max_{i\in[n]} \mathbb{E}_\mathbf{P} \sum_{j=i}^{i+\gamma D} 1_{\hat{s}_{[j]} \neq s_{[j]}}\right)$ | Exponentiated gradient $\times n$ | $\mathcal{O}\left(n\sqrt{T}\right)$ |
| Online job scheduling (5)* | Corrected offline optimal for predicted logits $\hat{\mathbf{x}} \in \mathbb{R}^m$ | Opt. weights $\mathbf{w} \in \triangle^m$ | $\mathcal{O}\left(\|\hat{\mathbf{x}} - \log \mathbf{w}\|_\infty\right)$ | Euclidean KT-OCO | $\mathcal{O}\left(\sqrt{mT\log(mT)}\right)$ |
| Non-clairvoyant job scheduling (6)‡ | Preferential round-robin with trade-off parameter $\lambda$ | Prediction quality $\eta$ | $\min\left\{\frac{1+2\eta/n}{1-\lambda}, \frac{2}{\lambda}\right\}$ | Exponential forecaster | $\mathcal{O}\left(\sqrt{T\log T}\right)$ |
| Ski-rental w. integer days $n \in [N]$ (6) | Buy if price $b \leq x$, $\lambda$ trade-off with worst-case approx. | Number of ski-days $n$ | $\frac{\min\{\lambda(b1_{x>b}+n1_{x<b}),b,n\}}{1-(1+1/b)^{-b\lambda}}$ | Exponentiated gradient | $\mathcal{O}\left(N\sqrt{T\log(NT)}\right)$ |
| Ski-rental with $\beta$-dispersed $n$ (6) | Buy after $x$ days, $\lambda$ trade-off with worst-case approx. | Number of ski-days $n$ | $\min\left\{\frac{e \min\{n,b\}}{(e-1)\lambda}, \frac{n1_{n\le x}+(b+x)1_{n>x}}{1-\lambda}\right\}$ | Exponential forecaster | $\mathcal{O}\left(\sqrt{T\log(NT)} + N^2T^{1-\beta}\right)$ |

* For these problems we also provide new guarantees in the statistical (i.i.d.) setting and for learning linear predictors that take instance features as their inputs.

† We also obtain results for its extensions to minimum-weight $\mathbf{b}$-matching and other graph algorithms with predictions in Appendices B and D.

‡ We also provide new guarantees for the problem of learning job permutations in the non-clairvoyant setting in Appendix E.

step approach inspired by recent work on theoretical meta-learning [25], which has been used to derive numerous multi-task learning results by optimizing regret-upper-bounds that encode the task-similarity [31, 32, 9, 26]. As evidenced by our results in Table 1, we believe the following two-step framework below holds similar potential for obtaining guarantees for algorithms with predictions:

1. For a given algorithm, derive a convenient-to-optimize upper bound $U_t(\mathbf{x})$ on the cost $C_t(\mathbf{x})$ that depends on both the prediction $\mathbf{x}$ and information specific to instance $t$ returned once the algorithm terminates, e.g. the optimum in combinatorial optimization. We find that in many cases such bounds already exist, and the quality of the prediction can be measured by a distance from some ground truth obtained from the output, a quantity that is usually convex and thus learnable.

2. Apply online learning to obtain both regret guarantees against adversarial sequences and sample complexity bounds for i.i.d. instances. We provide pseudo-code for a generic setup in Algorithm 1.

Table 1 summarizes instantiations of our framework on multiple problems. Our approach is designed to be simple-to-execute, leaving much of the difficulty to what the field is already good at: designing algorithms and proving prediction-quality-dependent upper bounds on their costs. Once the latter is accomplished, our framework leverages problem-specific structure to design a customized learning algorithm for each problem, leading to strong regret and sample complexity guarantees. In particular, in multiple settings we improve upon existing results in either sample complexity or generality, and in all cases we are the first to show regret guarantees in the online setting. This demonstrates the usefulness of and need for such a theoretical framework for studying these problems.

We summarize the diverse set of contributions enabled by our theoretical framework below:

1. **Bipartite matching:** Our starting example builds upon the work on **minimum-weight bipartite matching** using the Hungarian algorithm by Dinitz et al. [16]. We show how our framework leads directly to both the first regret guarantees in the online setting and new sample complexity bounds that improve over the previous approach by a factor linear in the number of nodes. In the Appendix we show similar strong improvements for $\mathbf{b}$-matching and other graph algorithms.

2. **Page migration:** We next study a more challenging application, **online page migration**, and show how we can adapt the algorithmic guarantee of Indyk et al. [22] into a learnable upper bound for which we can again provide both adversarial and statistical guarantees.

3. **Learning linear maps with instance-feature inputs:** Rather than assume the existence of a strong fixed prediction, it is often more natural to assume each instance comes with features that can be input into a predictor such as a linear map. Our approach yields the first guarantees for learning linear predictors for algorithms with predictions, which we obtain for the two problem settings above and also for **online job scheduling** using makespan minimization [30].

4. **Tuning robustness-consistency trade-offs:** Many bounds for *online* algorithms with predictions incorporate parameterized trade-offs between trusting the prediction or falling back on a worst-case approximation. This suggests the usefulness of tuning the trade-off parameter, which we instantiate on a simple job scheduling problem with a fixed predictor. Then we turn to the more challenging problem of simultaneously tuning the trade-off and learning predictions, which we achieve on two variants of the **ski-rental** problem. For the discrete case we give the only learning-theoretic guarantee, while for the continuous case our bound uses a dispersion assumption [6] that, in the i.i.d. setting, is a strictly weaker assumption than the log-concave requirement of Diakonikolas et al. [15].

## 2 Related work

Algorithms with predictions is a type of beyond-worst-case analysis of algorithms [42]; along with areas like smooth analysis [45] and data-driven algorithm design [3], it takes advantage of the fact that real-world instances are not worst-case. Inspired by success in applications such as learned indices [27], there has been a great deal of theoretical study focusing on algorithms whose guarantees depend on the quality of a given predictor (c.f. the Mitzenmacher and Vassilvitskii [36] survey). The actual learning of this predictor has been studied less [1, 15, 16] and rarely in the online setting; we aim to change this with our study. Some papers improve online learning itself using predictions [39, 23, 14], but they also assume known predictors or only learn over a small set of policies, and their goal is minimizing regret not computation. In-general, we focus on showing how algorithms with predictions can make use of online learning rather than on new methods for the latter. Several works [4, 40, 2] use learning *while* advising an algorithm, in-effect taking a learning-inspired approach to better make use of a prediction *within* an algorithm, whereas we focus on learning the prediction *outside* of the target algorithm. Our paper presents the first general framework for efficiently learning useful predictors.

Data-driven algorithm design is a related area that has seen more learning-theoretic effort [19, 6, 3]. At a high-level, it often studies tuning parameters such as the gradient descent step-size [19] or settings of branch and bound [5], whereas the predictors in algorithms with predictions guess the sequence in an online algorithm [22] or the actual outcome of the computation [16]. The distinction can be viewed as terminological, since a prediction can be viewed as a parameter, but it can mean that in our settings we have full information about the loss function since it is typically some discrepancy between the full sequence or computational outcome and the prediction. In contrast, in data-driven algorithm design getting the cost of each parameter often requires additional computation, leading to (semi-)bandit settings [7]. A more salient difference is that data-driven algorithm design guarantees compete with the parameter that minimizes average cost but do not always quantify the improvement attainable via learning; in algorithms with predictions we do generally quantify this improvement with an upper bound on the cost that depends on the prediction quality, but we usually only compete with the parameter that is optimal for prediction quality, which is not always cost-optimal. We do adapt data-driven algorithm design tools like dispersion [6] for algorithms with predictions.

Our two-step approach to providing guarantees for algorithms with predictions is inspired by the Average Regret-Upper-Bound Analysis (ARUBA) framework [25] for studying meta-learning [18]. Instead of instances they have tasks with different data-points, and the upper bounds are on learning-theoretic quantities such as regret rather than computational costs such as runtime. Mathematically, ARUBA takes advantage of similar structure in the regret-upper-bounds that we find in algorithms with predictions, namely that the upper bounds encode some measure of the quality of the prediction (in their case an initialization for gradient descent) via a comparison to the ground-truth (in their case the optimal parameter). However, whereas in ARUBA the need to know this optimal parameter after seeing a task is a weakness that does not hold in practice, in algorithms with predictions the corresponding quantity—the feedback listed in Table 1—is generally known after seeing the instance.

## 3 Framework overview and application to bipartite matching

In this section we outline the theoretical framework for designing algorithms and proving guarantees for learned predictors. As an illustrative example we will use the Hungarian algorithm for bipartite matching, for which Dinitz et al. [16] demonstrated an instance-dependent upper bound on the running time using a learned dual vector. Along the way, we will show an improvement to their sample complexity bound together with the first online results for this setting.

**Bipartite matching:** For a bipartite graph on $n$ nodes and $m$ edges, **min-weight perfect matching** (MWPM) asks for the perfect matching with the least weight according to edge-costs $\mathbf{c} \in \mathbb{Z}_{\geq 0}^m$. A common approach here is the Hungarian algorithm, a convex optimization-based approach for which Dinitz et al. [16] showed a runtime bound of $\tilde{\mathcal{O}}\left(m\sqrt{n}\min\left\{\|\mathbf{x} - \mathbf{x}^*(\mathbf{c})\|_1, \sqrt{n}\right\}\right)$, where $\mathbf{x} \in \mathbb{Z}^n$ initializes the duals in a primal-dual algorithm and $\mathbf{x}^*(\mathbf{c}) \in \mathbb{Z}^n$ is dual of the optimal solution; note that the latter is obtained for free after running the Hungarian method.

**Step 1 - Upper bound:** The first step of our approach is to find a suitable function $U_t(\mathbf{x})$ of the prediction $\mathbf{x}$ that (a) upper bounds the target algorithm's cost $C_t(\mathbf{x})$, (b) can be constructed completely once the algorithm terminates, and (c) can be efficiently optimized. These qualities allow learning the predictor in the second step. The requirements are similar to those of ARUBA for showing results for meta-learning [25], although there the quantity being upper-bounded was regret, not algorithmic cost.

**Algorithm 1:** Generic application of an online learning algorithm over $\mathcal{X}$ to learn a predictor for a method AlgorithmWithPrediction that takes advice from $\mathcal{X}$ and returns upper bounds $U_t$ on its cost. The goal of OnlineAlgorithm is low regret over the sequence $U_t$, so that on-average $C_t$ is upper bounded by the smallest possible average of $U_t$, up to some error decreasing in $T$. For specific instantiations of algorithms and feedback see Table 1.

---

initialize $\mathbf{x}_1 \in \mathcal{X}$ for OnlineAlgorithm
**for** *instance* $t = 1, \dots, T$ **do**
    obtain instance $I_t$
    run AlgorithmWithPrediction($I_t, \mathbf{x}_t$)
        suffer cost $C_t(\mathbf{x}_t) \le U_t(\mathbf{x}_t)$
        get feedback to construct upper bound $U_t$
    $\mathbf{x}_{t+1} \leftarrow$ OnlineAlgorithm($\{U_t\}_{s=1}^{t}, \mathbf{x}_1$)

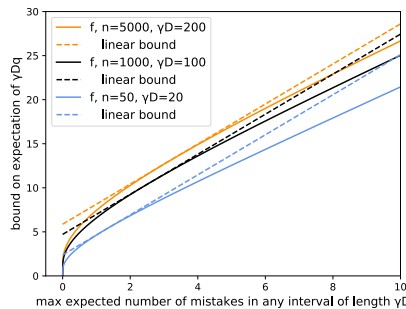

Figure 1: Bounds $f$ (c.f. Lemma 4.1) for three different $n$ and $\gamma D$ on the expected largest number of mistakes in any $\gamma D$-interval as a function of the maximum expected number $U_s(\mathbf{p})$ in any interval.

Many guarantees for algorithms with predictions are already amenable to being optimized, although we will see that they can require some massaging in order to be useful. In many cases the guarantee is a distance metric between the prediction $\mathbf{x}$ and some instance-dependent perfect prediction $\mathbf{x}^*$, which is convex and thus straightforward to learn. This is roughly true of our bipartite matching example, although taking the minimum of a constant and the distance $\|\mathbf{x} - \mathbf{x}^*(\mathbf{c})\|_1$ between the predicted and actual duals makes the problem non-convex. However, we can further upper bound their result by $\tilde{\mathcal{O}}\left(m\sqrt{n}\|\mathbf{x} - \mathbf{x}^*(\mathbf{c})\|_1\right)$; note that Dinitz et al. [16] also optimize this quantity, not the tighter upper bound with the minimum. While this might seem to be enough for step one, Dinitz et al. [16] also require the prediction $\mathbf{x}$ to be integral, which is difficult to combine with standard online procedures. In order to get around this issue, we show that rounding any nonnegative real vector to the closest integer vector incurs only a constant multiplicative loss in terms of the $\ell_1$-distance.

**Claim 3.1.** *Given any vectors $\mathbf{x} \in \mathbb{Z}^n$ and $\mathbf{y} \in \mathbb{R}^n$, let $\tilde{\mathbf{y}} \in \mathbb{Z}^n$ be the vector whose elements are those of $\mathbf{y}$ rounded to the nearest integer. Then $\|\mathbf{x} - \tilde{\mathbf{y}}\|_1 \le 2\|\mathbf{x} - \mathbf{y}\|_1$.*

*Proof.* Let $S \subset [n]$ be the set of indices $i \in [n]$ for which $\mathbf{x}_{[i]} \ge \mathbf{y}_{[i]} \iff \tilde{\mathbf{y}}_{[i]} = \lceil \mathbf{y}_{[i]} \rceil$. For $i \in [n]\backslash S$ we have $|\mathbf{x}_{[i]} - \mathbf{y}_{[i]}| \ge 1/2 \ge |\tilde{\mathbf{y}}_{[i]} - \mathbf{y}_{[i]}|$ so it follows by the triangle inequality that

$$\|\mathbf{x} - \tilde{\mathbf{y}}\|_1 = \sum_{i \in S} |\mathbf{x}_{[i]} - \tilde{\mathbf{y}}_{[i]}| + \sum_{i \in [n]\backslash S} |\mathbf{x}_{[i]} - \tilde{\mathbf{y}}_{[i]}| \le \sum_{i \in S} |\mathbf{x}_{[i]} - \mathbf{y}_{[i]}| + \sum_{i \in [n]\backslash S} |\mathbf{x}_{[i]} - \mathbf{y}_{[i]}| + |\mathbf{y}_{[i]} - \tilde{\mathbf{y}}_{[i]}|$$

$$\le \sum_{i \in S} |\mathbf{x}_{[i]} - \mathbf{y}_{[i]}| + 2 \sum_{i \in [n]\backslash S} |\mathbf{x}_{[i]} - \mathbf{y}_{[i]}| \le 2\|\mathbf{x} - \mathbf{y}\|_1$$

$\square$

Combining this projection with the convex relaxation above and the result of Dinitz et al. [16] shows that for any predictor $\mathbf{x} \in \mathbb{R}^n$ we have (up to affine transformation) a convex upper bound $U_t(\mathbf{x}) = \|\mathbf{x} - \mathbf{x}^*(\mathbf{c}_t)\|_1$ on the runtime of the Hungarian method, as desired. We now move to step two.

**Step 2 - Online learning:** Once one has an upper bound $U_t$ on the cost, the second component of our approach is to apply standard online learning algorithms and results to these upper bounds to obtain guarantees for learning predictions. In online learning, on each of a sequence of rounds $t = 1, \dots, T$ we predict $\mathbf{x}_t \in \mathcal{X}$ and suffer $U_t(\mathbf{x}_t)$ for some adversarially chosen loss function $U_t : X \mapsto \mathbb{R}$ that we then observe; the goal is to use this information to minimize **regret** $\sum_{t=1}^{T} U_t(\mathbf{x}_t) - \min_{\mathbf{x} \in \mathcal{X}} U_t(\mathbf{x})$, with the usual requirement being that it is **sublinear** in $T$ and thus decreasing on average over time. For bipartite matching, we can just apply regular **projected online (sub)gradient descent (OGD)** to losses $U_t(\mathbf{x}) = \|\mathbf{x} - \mathbf{x}^*(\mathbf{c}_t)\|_1$, i.e. the update rule $\mathbf{x}_{t+1} \leftarrow \arg\min_{\mathbf{x} \in \mathcal{X}} \alpha \langle \nabla U_t(\mathbf{x}_t), \mathbf{x} \rangle + \frac{1}{2}\|\mathbf{x}\|_2^2$ for appropriate step-size $\alpha > 0$; as shown in Theorem 3.1, this yields sublinear regret via a textbook result. The simplicity here is the point: by relegating as much of the difficulty as we can to obtaining an easy-to-optimize upper bound in step one, we make the actual learning-theoretic component easy. However, as we show in the following sections, it is not always easy to obtain a suitable upper bound, nor is it always obvious what online learning algorithm to apply, e.g. if the upper bounds are non-convex.

Our use of online learning is motivated by three factors: (1) doing well on non-i.i.d. instances is important in practical applications, e.g. in job scheduling where resource demand changes over time; (2) its extensive suite of algorithms lets us use different methods to tailor the approach to specific settings and obtain better bounds, as we exemplify via our use of exponentiated gradient over the simplex geometry in Section 4 and KT-OCO over unbounded Euclidean space in Section 5; (3) the existence of classic online-to-batch procedures for converting regret into sample complexity guarantees [12], i.e. bounds on the number of samples needed to obtain an $\varepsilon$-suboptimal predictor w.p. $\geq 1 - \delta$. While online-to-batch conversion can be suboptimal [20], as we show in Theorems 3.1, B.2, and D.1 its application to various graph algorithms with predictions problems *improves* upon existing sample complexity results. For completeness, we formalize online-to-batch conversion as Lemma A.1 in the Appendix.

We now show how to apply the second online learning step to bipartite matching by improving upon the result of Dinitz et al. [16] in Theorem 3.1; the improvement is the entirely new regret bound against adversarial cost vectors and a $\tilde{\mathcal{O}}(n)$ lower sample complexity. Note how the proof needs only their existing algorithmic contribution, Claim 3.1, and some standard tools in online convex optimization.

**Theorem 3.1.** *Suppose we have a fixed bipartite graph with $n \geq 3$ vertices and $m \geq 1$ edges.*

1. *For any cost vector $\mathbf{c} \in \mathbb{Z}_{\geq 0}^m$ and any dual vector $\mathbf{x} \in \mathbb{R}^n$ there exists an algorithm for MWPM that runs in time*
$$\tilde{\mathcal{O}}\left(m\sqrt{n}\min\left\{U(\mathbf{x}), \sqrt{n}\right\}\right) \leq \tilde{\mathcal{O}}\left(m\sqrt{n}U(\mathbf{x})\right)$$
*for $U(\mathbf{x}) = \|\mathbf{x} - \mathbf{x}^*(\mathbf{c})\|_1$, where $\mathbf{x}^*(\mathbf{c})$ the optimal dual vector associated with $\mathbf{c}$.*

2. *There exists a poly-time algorithm s.t. for any $\delta, \varepsilon > 0$ and distribution $\mathcal{D}$ over integer $m$-vectors with $\ell_\infty$-norm $\leq C$ it takes $\mathcal{O}\left(\left(\frac{Cn}{\varepsilon}\right)^2 \log\frac{1}{\delta}\right)$ samples from $\mathcal{D}$ and returns $\hat{\mathbf{x}}$ s.t. w.p. $\geq 1 - \delta$:*
$$\mathbb{E}_{\mathbf{c}\sim\mathcal{D}}\|\hat{\mathbf{x}} - \mathbf{x}^*(\mathbf{c})\|_1 \leq \min_{\|\mathbf{x}\|_\infty \leq C} \mathbb{E}_{\mathbf{c}\sim\mathcal{D}}\|\mathbf{x} - \mathbf{x}^*(\mathbf{c})\|_1 + \varepsilon$$

3. *Let $\mathbf{c}_1, \ldots, \mathbf{c}_T \in \mathbb{Z}_{\geq 0}^m$ be an adversarial sequence of $m$-vectors with $\ell_\infty$-norm $\leq C$. Then OGD with appropriate step-size has regret*
$$\max_{\|\mathbf{x}\|_\infty \leq C} \sum_{t=1}^{T} \|\mathbf{x}_t - \mathbf{x}^*(\mathbf{c}_t)\|_1 - \|\mathbf{x} - \mathbf{x}^*(\mathbf{c}_t)\|_1 \leq Cn\sqrt{2T}$$

*Proof.* The first result follows by combining Dinitz et al. [16, Theorem 13] with Claim 3.1. For the third result, let $\mathbf{x}_t$ be the sequence generated by running OGD [46] with step size $C/\sqrt{2T}$ on the losses $U_t(\mathbf{x}) = \|\mathbf{x} - \mathbf{x}^*(\mathbf{c}_t)\|_1$ over domain $[-C, C]^n$. Since these losses are $\sqrt{n}$-Lipschitz and the duals are $C\sqrt{n}$-bounded in Euclidean norm the regret guarantee follows from Shalev-Shwartz [44, Corollary 2.7]. For the second result, we apply standard online-to-batch conversion to the third result, i.e. we draw $T = \Omega\left(\left(\frac{Cn}{\varepsilon}\right)^2 \log\frac{1}{\delta}\right)$ samples $\mathbf{c}_t$, run OGD as above on the resulting losses $U_t$, and set $\hat{\mathbf{x}} = \frac{1}{T}\sum_{t=1}^{T} \mathbf{x}_t$ to be the average of the resulting predictions $\mathbf{x}_t$. The result follows by Lemma A.1. $\square$

This concludes an overview of our two-step approach for obtaining learning guarantees for algorithms with predictions. To summarize, we propose to (1) obtain simple-to-optimize upper bounds $U_t(\mathbf{x})$ on the cost of the target algorithm on instance $t$ as a function of prediction $\mathbf{x}$ and (2) optimize $U_t(\mathbf{x})$ using online learning. While conceptually simple, even in this illustrative example it already improves upon past work; in the sequel we demonstrate further results that this approach makes possible. Note that, like Dinitz et al. [16], we are also able to generalize Theorem 3.1 to $\mathbf{b}$-matchings, which we do in Appendix B; another advantage of our approach is that it lets us prove online and statistical learning in the case where the demand vector $\mathbf{b}$ varies across instances rather than staying fixed as in Dinitz et al. [16]. Finally, in Appendix D we also improve upon the more recent learning-theoretic results of Chen et al. [13] for related graph algorithms with predictions problems.

## 4   Predicting requests for page migration

Equipped with our two-step approach for deriving guarantees for learning predictors, we investigate several more important problems in combinatorial optimization, starting with the page migration problem. Our results demonstrate that even for learning such simple predictors there are interesting technical challenges in deriving a *learnable* upper bound. Nevertheless, once this is accomplished the second step of our approach is again straightforward.

**Page migration:** Consider a server that sees a sequence of requests $s_{[1]}, \dots, s_{[n]}$ from metric space $(\mathcal{K}, d)$ and at each timestep decides whether to change its state $a_{[i]} \in \mathcal{K}$ at cost $Dd(a_{[i-1]}, a_{[i]})$ for some $D > 1$; it then suffers a further cost $d(a_{[i]}, s_{[i]})$. The **online page migration** (OPM) problem is then to minimize the cost to the server. Recently, Indyk et al. [22] studied a setting where we are given a sequence of predicted points $\hat{s}_{[1]}, \dots, \hat{s}_{[n]} \in \mathcal{K}$ to aid the page migration algorithm. They show that if there exists $\gamma, q \in (0, 1)$ s.t. $\gamma D \in [n]$ and for any $i \in [n]$ we have $\sum_{j=i}^{i+\gamma D - 1} 1_{s_{[j]} \neq \hat{s}_{[j]}} \leq q\gamma D$ then there exists an algorithm with competitive ratio $(1 + \gamma)(1 + \mathcal{O}(q))$ w.r.t. to the offline optimal. This algorithm depends on $\gamma$ but not $q$, so we study the setting where $\gamma$ is fixed.

**Deriving an upper bound:** As in the previous section, the predictions are discrete, so to use our approach we must convert it into a continuous problem. As we have fixed $\gamma$, the competitive ratio is an affine function of the following upper bound on $q$:

$$Q(\hat{s}, s) = \frac{1}{\gamma D} \max_{i \in [n - \gamma D + 1]} \sum_{j=i}^{i+\gamma D - 1} 1_{\hat{s}_{[j]} \neq s_{[j]}}$$

We assume that the set of points $\mathcal{K}$ is finite with indexing $k = 1, \dots, |\mathcal{K}|$ and use this to introduce our continuous relaxation, a natural randomized approach converting the problem of learning a prediction into $n$ experts problems on $|\mathcal{K}|$ experts. For each $j \in [n]$ we define a probability vector $\mathbf{p}_{[j]} \in \triangle_{|\mathcal{K}|}$ governing the categorical r.v. $\hat{s}_{[j]}$, i.e. $\Pr\{\hat{s}_{[j]} = k\} = \mathbf{p}_{[j,k]} \; \forall \, k \in \mathcal{K}$. Under these distributions the expected competitive ratio will be $(1 + \gamma)(1 + \mathcal{O}(\mathbb{E}_{\hat{s} \sim \mathbf{p}} Q(\hat{s}, s)))$, for $\mathbf{p}$ the product distribution of the vectors $\mathbf{p}_j$. Note that forcing each $\mathbf{p}_j$ to be a one-hot vector recovers the original approach with no loss, so optimizing $\mathbb{E}_{\hat{s} \sim \mathbf{p}} Q(\hat{s}, s)$ over $\mathbf{p} \in \triangle_{|\mathcal{K}|}^n$ would find a predictor that fits the original result.

However, $\mathbb{E}_{\hat{s} \sim \mathbf{p}} Q(\hat{s}, s)$ is not convex in $\mathbf{p}$. The simplest relaxation is to replace the maximum by summation, but this leads to a worst-case bound of $\mathcal{O}\left(\frac{n}{\gamma D}\right)$. We instead bound $\mathbb{E}_{\hat{s} \sim \mathbf{p}} Q(\hat{s}, s)$—and thus also the expected competitive ratio—by a function of the following maximum over expectations:

$$U_s(\mathbf{p}) = \max_{i \in [n - \gamma D + 1]} \mathbb{E}_{\hat{s} \sim \mathbf{p}} \sum_{j=i}^{i+\gamma D - 1} 1_{\hat{s}_{[j]} \neq s_{[j]}} = \max_{i \in [n - \gamma D + 1]} \sum_{j=i}^{i+\gamma D - 1} 1 - \langle \mathbf{s}_{[j]}, \mathbf{p}_{[j]} \rangle$$

where $\mathbf{s}_{[j,k]} = 1_{s_{[j]} = k} \; \forall \, k \in \mathcal{K}$, i.e. $\mathbf{s}_{[j]}$ encodes the location in $\mathcal{K}$ of the $j$th request. As a maximum over $n - \gamma D + 1$ convex functions this objective is also convex. Note also that if $U_s(\mathbf{p})$ is zero—i.e. the probability vectors are one-hot and perfect—then $\mathbb{E}_{\hat{s} \sim \mathbf{p}} Q(\hat{s}, s) \geq q$ will also be zero. In fact, $q$ is upper-bounded by a monotonically increasing function of $U_s(\mathbf{p})$ that is zero at the origin, but as this function is concave and non-Lipschitz (c.f. Figure 1) we incur an additive $\mathcal{O}\left(\frac{\log(n - \gamma D + 1)}{\gamma D}\right)$ loss to obtain an online-learnable upper bound. This is formalized in the following result (proof in A.1).

**Lemma 4.1.** *There exist constants $a < e, b < 2/e$ and a monotonically increasing function $f : [0, \infty) \mapsto [0, \infty)$ s.t. $f(0) = 0$ and*

$$\mathbb{E}_{\hat{s} \sim \mathbf{p}} Q(\hat{s}, s) \leq \frac{f(U_s(\mathbf{p}))}{\gamma D} \leq \frac{aU_s(\mathbf{p}) + b\log(n - \gamma D + 1)}{\gamma D}$$

We now have an convex bound on the competitive ratio for the OPM algorithm of Indyk et al. [22]. For both this and bipartite matching we resorted to a relaxation of a discrete problem. However, whereas before we only incurred a multiplicative loss (c.f. Claim 3.1), here we have an additive loss that makes the bound meaningful only for $\gamma D \gg \log n$. However, as the method we propose optimizes $U_s(\mathbf{p})$, which bounds $q$ with no additive error via the function $f$ in Lemma 4.1, in-practice we may expect it to help minimize $q$ in all regimes. Note that the non-Lipschitzness near zero that prevents using $f$ for formal regret guarantees comes from the poor tail behavior of Poisson-like random variables with small means, which we do not expect can be significantly improved.

**Learning guarantees:** Having established an upper bound, in Theorem 4.2 we again show how a learning-theoretic result follows from standard online learning. This time, instead of OGD we run **exponentiated (sub)gradient (EG)** [44], a classic method for learning from experts, on each of $n$ simplices to learn the probabilities $\mathbf{p}_{[j]} \; \forall \, j \in [n]$. The multiplicative update $\mathbf{x}_{t+1} \propto \mathbf{x}_t \odot \exp(-\alpha \nabla U_t(\mathbf{x}_t))$ of EG is notable for yielding regret logarithmic in the size $|\mathcal{K}|$ of the simplices, which is important for large metric spaces. Note that as the relaxation is randomized, our algorithms output a dense probability vector; to obtain a prediction for OPM we sample $\hat{s}_{t[j]} \sim \mathbf{p}_{[j]} \; \forall \, j \in [n]$.

**Theorem 4.2.** *Let $(\mathcal{K}, d)$ be a finite metric space.*

1. *For any request sequence $s$ and any set of probability vectors $\mathbf{p} \in \triangle_{|\mathcal{K}|}^n$ there exists an algorithm for OPM with expected competitive ratio*

$$(1 + \gamma)\left(1 + \mathcal{O}\left(\frac{U_s(\mathbf{p}) + \log(n - \gamma D + 1)}{\gamma D}\right)\right)$$

2. *There exits a poly-time algorithm s.t. for any $\delta, \varepsilon > 0$ and distribution $\mathcal{D}$ over request sequences $s \in \mathcal{K}^n$ it takes $\mathcal{O}\left(\left(\frac{\gamma D}{\varepsilon}\right)^2\left(n^2 \log |\mathcal{K}| + \log \frac{1}{\delta}\right)\right)$ samples from $\mathcal{D}$ and returns $\hat{\mathbf{p}}$ s.t. w.p. $\geq 1 - \delta$:*

$$\mathbb{E}_{s \sim \mathcal{D}} U_s(\hat{\mathbf{p}}) \leq \min_{\mathbf{p} \in \triangle_{|\mathcal{K}|}^n} \mathbb{E}_{s \sim \mathcal{D}} U_s(\mathbf{p}) + \varepsilon$$

3. *Let $s_1, \ldots, s_T$ be an adversarial sequence of request sequences. Then updating the distribution $\mathbf{p}_{t[j]}$ over $\triangle_{|\mathcal{K}|}$ at each timestep $j \in [n]$ using EG with appropriate step-size has regret*

$$\max_{\mathbf{p} \in \triangle_{|\mathcal{K}|}^n} \sum_{t=1}^T U_{s_t}(\mathbf{p}_t) - U_{s_t}(\mathbf{p}) \leq \gamma Dn\sqrt{2T \log |\mathcal{K}|}$$

*Proof.* The first result follows by combining Indyk et al. [22, Theorem 1] with Lemma 4.1. For the third let $\mathbf{p}_t$ be generated by running $n$ exponentiated gradient algorithms with step-size $\sqrt{\frac{\log |\mathcal{K}|}{2\gamma^2 D^2 T}}$ on losses $U_{s_t}(\mathbf{p})$ over $\triangle_{|\mathcal{K}|}^n$. Since these are $\gamma D$-Lipschitz and the maximum entropy is $\log |\mathcal{K}|$, the regret follows by [44, Theorem 2.15]. For the second result, apply standard online-to-batch conversion to the third, i.e. draw $T = \Omega\left(\left(\frac{\gamma D}{\varepsilon}\right)^2\left(n^2 \log |\mathcal{K}| + \log \frac{1}{\delta}\right)\right)$ samples $\mathbf{s}_t$, run EG on $U_{s_t}(\mathbf{p})$ as above, and set $\hat{\mathbf{p}} = \frac{1}{T}\sum_{t=1}^T \mathbf{p}_t$ to be the average of the resulting actions. The result follows by Lemma A.1. $\square$

As before, this result first shows how the quantity of interest—here the competitive ratio—is upper-bounded by an affine function of some quality measure $U_s(\mathbf{p})$, for which we then provide regret and statistical guarantees using online learning. The difficulty deriving a suitable bound exemplifies the technical challenges that arise in learning predictors, and may also be encountered in other sequence prediction problems such as TCP [10]. Nevertheless, our approach does yield an online procedure that incurs only $\mathcal{O}(\frac{\log n}{\gamma D})$ additive error over Indyk et al. [22] in the case of a perfect predictor and, unlike their work, we provide an algorithm for learning the predictor itself. In Appendix C.2 we also show an auto-regressive extension which does *not* require learning a distribution for each timestep $j \in [n]$.

## 5 Learning linear predictors with instance-feature inputs

So far we have considered only *fixed* predictors, either optima-in-hindsight in the online setting or a population risk minimizers for i.i.d. data. Actual instances can vary significantly and so a fixed predictor may not be very good, e.g. in the example of querying a sorted array it means always returning the same index. In the online setting one can consider methods that adapt to dynamic comparators [46, 23, 37], which are also applicable to our upper bounds; however, these still need measures such as the comparator path-length to be small, which may be more reasonable in some cases but not all.

We instead study the setting where all instances come with instance-specific features, a natural and practical assumption [27, 30] that encompasses numerical representations of the instance itself—e.g. bits representing a query or a graph—or other information such as weather or day of the week. These are passed to functions—e.g. linear predictors, neural nets, or trees—whose parameters can be learned from data. We study linear predictors, which are often amenable to similar analyses as above since the composition of a convex and affine function is convex. For example, it is straightforward to extend the matching results to learning linear predictors of duals. OPM is more challenging because the outputs must lie in the simplex, which can be solved by learning rectangular stochastic matrices. Both sets of results are shown in Appendix C. Notably, for page migration our guarantees cover the auto-regressive setting where the server probabilities are determined by a fixed linear transform of past states.

Our main example will be online job scheduling via minimizing the fractional makespan [30], where we must assign each in a sequence of variable-sized jobs to one of $m$ machines. Lattanzi et al. [30] provide an algorithm that uses predictions $\hat{\mathbf{w}} \in \mathbb{R}_{>0}^m$ of "good" machine weights $\mathbf{w} \in \mathbb{R}_{>0}^m$ to assign jobs

based on how well $\hat{\mathbf{w}}$ corresponds to machine demand; the method has a performance guarantee of $\mathcal{O}(\log \min\{\max_i \frac{\hat{\mathbf{w}}_{[i]}}{\mathbf{w}_{[i]}}, m\})$. They also discuss learning linear and other predictors, but without guarantees. We study linear prediction of the *logarithm* of the machine weights, which makes the problem convex, and assume features lie in the $f$-dimensional simplex. For simplicity we only consider learning the linear transform from features to predictors and not the intercept, as the former subsumes the latter. For the online result, we use **KT-OCO** [38, Algorithm 1], a parameter-free subgradient method with update $\mathbf{x}_{t+1} \leftarrow \frac{1+\sum_{s=1}^{t}\langle \mathbf{g}_s, \mathbf{x}_s \rangle}{t+1} \sum_{s=1}^{t} \mathbf{g}_s$ for $\mathbf{g}_s = \nabla U_s(\mathbf{x}_s)$; it allows us to not assume any bound on the machine weights and thus to compete with the optimal linear predictor in all of $\mathbb{R}^{m \times f}$.

**Theorem 5.1.** *Consider online restricted assignment with $m \geq 1$ machines [30, Section 2.1].*

1. *For predicted logits $\mathbf{x} \in \mathbb{R}^m$ there is an algorithm whose fractional makespan has competitive ratio*

$$\mathcal{O}(\min\{\|\mathbf{x} - \log \mathbf{w}\|_\infty, \log m\}) \leq \mathcal{O}(U(\mathbf{x}))$$

   *for $U(\mathbf{x}) = \|\mathbf{x} - \log \mathbf{w}\|_\infty$, where $\mathbf{w} \in \mathbb{R}_{>0}^m$ are good machine weights [30, Section 3].*
2. *There exists a poly-time algorithm s.t. for any $\delta, \varepsilon > 0$ and distribution $\mathcal{D}$ over machine (weight, feature) pairs $(\mathbf{w}, \mathbf{f}) \in \mathbb{R}_{>0}^m \times \triangle_f$ s.t. $\|\log \mathbf{w}\|_\infty \leq B$ the algorithm takes $\mathcal{O}\left(\left(\frac{B}{\varepsilon}\right)^2 \left(mf + \log \frac{1}{\delta}\right)\right)$ samples from $\mathcal{D}$ and returns $\hat{\mathbf{A}} \in \mathbb{R}^{m \times f}$ s.t. w.p. $\geq 1 - \delta$*

$$\mathbb{E}_{(\mathbf{w}, \mathbf{f}) \sim \mathcal{D}}\|\hat{\mathbf{A}}\mathbf{f} - \log \mathbf{w}\|_\infty \leq \min_{\|\mathbf{A}\|_{\max} \leq B} \mathbb{E}_{(\mathbf{w}, \mathbf{f}) \sim \mathcal{D}}\|\mathbf{A}\mathbf{f} - \log \mathbf{w}\|_\infty + \varepsilon$$

3. *Let $(\mathbf{w}_1, \mathbf{f}_1), \ldots, (\mathbf{w}_T, \mathbf{f}_T) \in \mathbb{R}_{>0}^m \times \triangle_f$ be an adversarial sequence of (weights, feature) pairs. Then for any $\mathbf{A} \in \mathbb{R}^{m \times f}$ KT-OCO has regret*

$$\sum_{t=1}^{T} \|\mathbf{A}_t \mathbf{f}_t - \log \mathbf{w}_t\|_\infty - \|\mathbf{A}\mathbf{f}_t - \log \mathbf{w}_t\|_\infty \leq \|\mathbf{A}\|_F \sqrt{T \log(1 + 24T^2 \|\mathbf{A}\|_F^2)} + 1$$

   *If we restrict to matrices with entries bounded by $B$ then OGD with appropriate step-size has regret*

$$\max_{\|\mathbf{A}\|_{\max} \leq B} \sum_{t=1}^{T} \|\mathbf{A}_t \mathbf{f}_t - \log \mathbf{w}_t\|_\infty - \|\mathbf{A}\mathbf{f}_t - \log \mathbf{w}_t\|_\infty \leq B\sqrt{2mfT}$$

*Proof.* The first result follows by substituting $\max_i \frac{\exp(\mathbf{x}_{[i]})}{\mathbf{w}_{[i]}}$ for $\eta$ in Lattanzi et al. [30, Theorem 3.1] and upper bounding the maximum by the $\ell_\infty$-norm. For the third, since $U_t$ is 1-Lipschitz w.r.t. the Euclidean norm we apply the guarantee for KT-OCO [38, Algorithm 1] using $\varepsilon = 1$ and the subgradients of $\|\mathbf{A}_t \mathbf{f}_t - \log \mathbf{w}_t\|_\infty$ as rewards [38, Corollary 5]. The result for $B$-bounded $\mathbf{A}$ follows by applying OGD with step-size $B\sqrt{\frac{mf}{2T}}$ over $\|\mathbf{A}\|_{\max} \leq B$ [44, Corollary 2.7]. Finally, the second result follows by applying online-to-batch conversion to the latter result, i.e. draw $T = \Omega\left(\left(\frac{B}{\varepsilon}\right)^2 \left(mf + \log \frac{1}{\delta}\right)\right)$ samples $(\mathbf{w}_t, \mathbf{f}_t)$, run OGD on the resulting losses $\|\mathbf{A}\mathbf{f} - \log \mathbf{w}_t\|_\infty$ as above, and set $\hat{\mathbf{A}} = \frac{1}{T}\sum_{t=1}^{T} \mathbf{A}_t$ to be the average of the resulting actions $\mathbf{A}_t$. The result follows by Lemma A.1. $\square$

This guarantee is the first we are aware of for learning non-static predictors in the algorithms with predictions literature. It demonstrates both how to extend fixed predictor results to learning linear predictors—note that the former is recovered by having $\mathbf{f}_t = \mathbf{1}_1 \ \forall \ t$—and how to handle unbounded predictor domains. The ability to provide such guarantees is another advantage of our approach.

## 6   Tuning robustness-consistency trade-offs for scheduling and ski-rental

We turn to tuning robustness-consistency trade-offs, introduced in Lykouris and Vassilvitskii [34]. This trade-off captures the tension between following the predictions when they are good (consistency) and doing not much worse than the worst-case guarantee in either case (robustness). In many cases, this trade-off can be made explicit, by a parameter $\lambda \in [0, 1]$. The setting of $\lambda$ is crucial, yet previous work left the decision to the end-user. Here we show that it is often eminently learnable in an online setting. We then demonstrate how to accomplish a much harder task—tuning $\lambda$ at the same time as learning to predict—on two related but technically very different variants of the ski-rental problem. This meta-application highlights the applicability of our approach to non-convex upper bounds.

**Robustness-consistency trade-offs:** Most problems studied in online algorithms with predictions usually have existing worst-case guarantees on the competitive ratio, i.e. a constant $\gamma \geq 1$ on how much worse (multiplicatively) a learning-free algorithm does relative to the offline optimal cost $\text{OPT}_t$ on instance $t$. While the goal of algorithms with predictions is to use data to do better than this worst-case bound, an imperfect prediction may lead to much worse performance. As a result, most guarantees strive to upper bound the cost of an algorithm with prediction $\mathbf{x}$ on an instance $t$ as follows:

$$C_t(\mathbf{x}, \lambda) \leq \min\left\{f(\lambda)u_t(\mathbf{x}), g_t(\lambda)\right\}$$

Here $u_t$ is some measure of the quality of $\mathbf{x}$ on instance $t$, $\lambda \in [0, 1]$ is a parameter, $f$ is a monotonically increasing function that ideally satisfies $f(0) = 1$, and $g_t$ is a monotonically decreasing function that ideally satisfies $g_t(1) = \gamma\text{OPT}_t$. A very common structure is $f(\lambda) = 1/(1 - \lambda)$ and $g_t(\lambda) = \frac{\gamma}{\lambda}\text{OPT}_t$. For example, consider job scheduling with predictions, a setting where we are given $n$ jobs and their predicted runtimes with total absolute error $\eta$ and must minimize the sum of their completion times when running on a single server with pre-emption. Here Kumar et al. [29, Theorem 3.3] showed that a preferential round-robin algorithm has competitive ratio at most $\min\left\{\frac{1+2\eta/n}{1-\lambda}, \frac{2}{\lambda}\right\}$. Thus if we know the prediction is perfect we can set $\lambda = 0$ and obtain the optimal cost (consistency); on the other hand, if we know the prediction is poor we can set $\lambda = 1$ and get the (tight) worst-case guarantee of two (robustness).

Of course in-practice we often do not know how good a prediction is on a specific instance $t$; we thus would like to learn to set $\lambda$, i.e. to learn how trustworthy our prediction is. As a first step, we can consider doing so when we are given a prediction for each instance and thus only need to optimize over $\lambda$. For example, the just-discussed problem of job scheduling has competitive ratio upper-bounded by $U_t(\lambda) = \min\left\{\frac{1+2\eta_t/n_t}{1-\lambda}, \frac{2}{\lambda}\right\}$ for $n_t$ and $\eta_t$ the number of jobs and the prediction quality, respectively, on instance $t$. Assuming a bound $B$ on the average error makes $U_t$ Lipschitz, so we can apply the **exponentially weighted average forecaster** [28, Algorithm 1], also known as the **exponential forecaster**. This algorithm, whose action at each time $t+1$ is to sample from the distribution with density $\rho_{t+1}(\cdot) \propto \rho_1(\cdot)\exp(-\alpha\sum_{s=1}^{t}U_s(\cdot))$, has the following regret guarantee (proof in A.2):

**Corollary 6.1.** *For the competitive ratio upper bounds $U_t$ of the job scheduling problem with average prediction error $\eta/n_t$ at most $B$ the exponential forecaster with appropriate step-size has expected regret*

$$\max_{\lambda\in[0,1]} \mathbb{E}\sum_{t=1}^{T} U_t(\lambda_t) - U_t(\lambda) \leq 9B\left(1 + \sqrt{\frac{T}{2}\log T}\right)$$

Thus a standard learning method produces a sequence $\lambda_t$ that performs as well as the best $\lambda$ asymptotically. We next consider the more difficult problem of simultaneously tuning $\lambda$ learning to predict.

**Ski-rental:** We instantiate this challenge on ski-rental, in which each task $t$ is a ski season with an unknown number of days $n_t \in \mathbb{Z}_{\geq 2}$; to ski each day, we must either buy skis at price $b_t$ or rent each day for the price of one. The optimal offline behavior is to buy iff $b_t < n_t$, and the best algorithm has worst-case competitive ratio $e/(e-1)$. Kumar et al. [29] and Bamas et al. [10, Theorem 2] further derive an algorithm with the following robustness-consistency trade-off between blindly following a prediction $x$ and incurring cost $u_t(x) = b_t\mathbb{1}_{x>b_t} + n_t\mathbb{1}_{x\leq b_t}$ or going with the worst-case guarantee:

$$U_t(x, \lambda) = \frac{\min\{\lambda u_t(x), b_t, n_t\}}{1 - e_t(-\lambda)}, e_t(z) = (1 + 1/b_t)^{b_t z}$$

Assuming a bound of $N \geq 2$ on the number of days and $B > 0$ on the buy price implies that $U_t$ is bounded and Lipschitz w.r.t. $\lambda$. We can thus run exponentiated gradient on the functions $U_t$ to learn a categorical distribution over the product set $[N] \times \{\delta/2, \ldots, 1 - \delta/2\}$ for some $\delta$ s.t. $1/\delta \in \mathbb{Z}_{\geq 2}$. This yields the following bound on the expected regret (proof in A.3).

**Corollary 6.2.** *For the competitive ratio upper bounds $U_t$ of the discrete ski-rental problem the randomized exponentiated gradient algorithm with an appropriate step-size has expected regret*

$$\max_{x\in[N],\lambda\in(0,1]} \mathbb{E}\sum_{t=1}^{T} U_t(x_t, \lambda_t) - U_t(x, \lambda) \leq 6N\sqrt{T\log(BNT)}$$

Thus via an appropriate discretization the sequence of predictions $(x_t, \lambda_t)$ does as well as the joint optimum on this problem. However, we can also look at a case where we are not able to just discretize

to get low regret. In particular, we consider the *continuous* ski-rental problem, where each day $n_t > 1$ is a real number, and study how to pick thresholds $x$ after which to buy skis, which has cost $u_t(x) = n_t 1_{n_t \leq x} + (b_t + x) 1_{n_t > x}$. Note that $x = 0$ and $x = N$ recovers the previous setting where our decision was to buy or not at the beginning. For this setting, Diakonikolas et al. [15] adapt an algorithm of Mahdian et al. [35] to bound the cost as follows:

$$C_t(x, \lambda) \leq U_t(x, \lambda) = \min \left\{ \frac{u_t(x)}{1 - \lambda}, \frac{e \min\{n_t, b_t\}}{(e - 1)\lambda} \right\}$$

While the bound is simpler as a function of $\lambda$, it is discontinuous in $x$ because $u_t$ is piecewise-Lipschitz. Since one cannot even attain sublinear regret on adversarially chosen threshold functions, we must make an assumption on the data. In particular, we will assume the days are *dispersed*:

**Definition 6.3.** A set of (possibly random) points $n_1, \ldots, n_T \in \mathbb{R}$ are $\beta$-**dispersed** if $\forall \, \varepsilon \geq T^{-\beta}$ the expected number in any $\varepsilon$-ball is $\tilde{\mathcal{O}}(\varepsilon T)$, i.e. $\mathbb{E} \max_{x \in [0, N]} |[x \pm \varepsilon] \cap \{n_1, \ldots, n_T\}| = \tilde{\mathcal{O}}(\varepsilon T)$.

Dispersion encodes the stipulation that the days, and thus the discontinuities of $u_t(x, \lambda)$, are not too concentrated. In the i.i.d. setting, a simple condition that leads to dispersion with $\beta = 1/2$ is the assumption that the points are drawn from a $\kappa$-bounded distribution [6, Lemma 1]. Notably this is a strictly weaker assumption than the log-concave requirement of Diakonikolas et al. [15] that they used to show statistical learning results for ski-rental. Having stipulated that the ski-days are $\beta$-dispersed, we can show that it implies dispersion of the loss functions [6] and thus obtain the following guarantee for the exponential forecaster applied to $U_t(x, \lambda)$ (proof in A.4):

**Corollary 6.4.** *For cost upper bounds $U_t$ of the continuous ski-rental problem the exponential forecaster with an appropriate step-size has expected regret*

$$\max_{x \in [0, N], \lambda \in (0, 1]} \mathbb{E} \sum_{t=1}^{T} U_t(x_t, \lambda_t) - U_t(x, \lambda) \leq \tilde{\mathcal{O}} \left( \sqrt{T \log(NT)} + (N + B)^2 T^{1 - \beta} \right)$$

Thus in two mathematically quite different settings of ski-rental we can directly apply online learning to existing bounds to not only learn online the best action for ski-rental, but to at the same time learn how trustworthy the best action is via tuning the robustness-consistency trade-off.

## 7 Conclusion and future work

The field of algorithms with predictions has been successful in circumventing worst case lower bounds and showing how simple predictions can improve algorithm performance. However, except for a few problem-specific approaches, the question of *how* to predict has largely been missing from the discussion. In this work we presented the first general framework for efficiently learning useful predictions and applied it to a diverse set of previously studied problems, giving the first low regret learning algorithms, reducing sample complexity bounds, and showing how to learn the best consistency-robustness trade-off. One current limitation is the lack of more general-case guarantees for *simultaneously* tuning robustness-consistency and learning the predictor, which we only show for ski-rental. There are also several other avenues for future work. The first is to build on our results and provide learning guarantees for other problems where the algorithmic question of *how* to use predictions is already addressed. Another is to try to improve known bounds by solving the problems holistically: developing easy-to-learn parameters in concert with developing algorithms that can use them. Finally, there is the direction of identifying hard problems: what are the instances where no reasonable prediction can help improve an algorithm's performance?

## Acknowledgments

We thank Yilin Yan, Alexander Smola, Shinsaku Sakaue, and Taihei Oki for helpful discussion. This material is based on work supported in part by the National Science Foundation under grants CCF-1535967, CCF-1910321, IIS-1618714, IIS-1705121, IIS-1838017, IIS-1901403, IIS-2046613, and SES-1919453; the Defense Advanced Research Projects Agency under cooperative agreements HR00112020003 and FA875017C0141; a Simons Investigator Award; an AWS Machine Learning Research Award; an Amazon Research Award; a Bloomberg Research Grant; a Microsoft Research Faculty Fellowship; an Amazon Web Services Award; a Facebook Faculty Research Award; funding from Booz Allen Hamilton Inc.; a Block Center Grant; and a Facebook PhD Fellowship. Any opinions, findings and conclusions or recommendations expressed in this material are those of the author(s) and do not necessarily reflect the views of any of these funding agencies.

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
