# A Proofs of main results

## A.1 Proof of Lemma 4.1

*Proof.* For each $j \in [n]$ define $p_j = 1 - \langle \mathbf{s}_{[j]}, \mathbf{p}_{[j]} \rangle$, i.e. the probability that $\hat{s}_j \neq s_j$, and the r.v. $X_j \sim \mathrm{Ber}(p_j)$. Define also the r.v. $S_i = \sum_{j=i}^{i+\gamma D-1} X_j$, s.t. we have

$$\gamma D \mathbb{E}_{\mathbf{p}} q = \mathbb{E}_{\mathbf{p}} \max_{i \in [n-\gamma D+1]} S_i = \mathbb{E}_{\mathbf{p}} \max_{i \in [n-\gamma D+1]} \sum_{j=i}^{i+\gamma D-1} X_j$$

Note that $S_i$ is a Poisson binomial and so has moment-generating function $\mathbb{E}_{\mathbf{p}} \exp(tS_i) = \prod_{j=i}^{i+\gamma D-1}(1 - p_j + p_j e^t)$. Therefore applying Jensen's inequality and the union bound yields

$$\exp\left( t \mathbb{E}_{\mathbf{p}} \max_{i \in [n-\gamma D+1]} S_i \right) \leq \mathbb{E}_{\mathbf{p}} \exp\left( t \max_{i \in [n-\gamma D+1]} S_i \right) = \mathbb{E}_{\mathbf{p}} \max_{i \in [n-\gamma D+1]} \exp\left( tS_i \right)$$

$$\leq \sum_{i=1}^{n-\gamma D+1} \mathbb{E}_{\mathbf{p}} \exp\left( tS_i \right)$$

$$\leq (n - \gamma D + 1) \prod_{j=i^*}^{i^*+\gamma D-1} (1 - p_j + p_j e^t)$$

for all $t > 0$ and $i^* \in \arg\max_{i \in [n-\gamma D+1]} \mathbb{E}_{\mathbf{p}} S_i$. We then have

$$t \mathbb{E}_{\mathbf{p}} \max_{i \in [n-\gamma D+1]} S_i \leq \log(n - \gamma D + 1) + \sum_{j=i^*}^{i^*+\gamma D-1} \log(1 - p_j + p_j e^t)$$

$$\leq \log(n - \gamma D + 1) + \sum_{j=i^*}^{i^*+\gamma D-1} \log \exp(p_j(e^t - 1))$$

$$\leq \log(n - \gamma D + 1) + \sum_{j=i^*}^{i^*+\gamma D-1} p_j(e^t - 1)$$

$$\leq \log(n - \gamma D + 1) + \mathbb{E}_{\mathbf{p}} S_{i^*}(e^t - 1)$$

Dividing by $t = W\left( \frac{\log(n-\gamma D+1)}{x} \right) + 1$ shows that $f(x) = \frac{x(\exp(t)-1)+\log(n-\gamma D+1)}{t\gamma D}$, where $W : [0, \infty) \mapsto [0, \infty)$ is the Lambert $W$-function. Define $L = \log(n - \gamma D + 1)/e$, so we are interested in bounding $f(x) = \frac{x(\exp(W(L/x)+1)-1)+eL}{W(L/x)+1}$. We compute its derivative:

$$f'(x) = \frac{x(\exp(W(L/x) + 1) - 1)W(L/x)^2 - 3x(W(L/x) + 1/3) + x\exp(W(L/x) + 1) + 2eL}{x(W(L/x) + 1)^3}$$

and second derivative:

$$f''(x) = -\frac{W(L/x)\left((x + eL)W(L/x)^2 + 2(2x + eL)W(L/x) + eL\right)}{x^2(W(L/x) + 1)^5}$$

Since the second derivative is always negative, $f$ is a concave function on $x \geq 0$. Thus for $\omega = W(1)$ we have

$$f(x) \leq \min_{y>0} f(y) + f'(y)(x - y)$$

$$\leq \frac{L(e/\omega - 1 + e)}{\omega + 1} + \frac{L(e/\omega - 1)\omega^2 - 3L(\omega + 1/3) + Le/\omega + 2eL}{L(\omega + 1)^3}(x - L)$$

$$= \left( e/\omega + 1/(\omega + 1)^3 - (e + 1)/(\omega + 1) - 1/(\omega + 1)^2 \right) x$$

$$\quad + \left( 1/(\omega + 1)^2 + e/(\omega + 1) - 1/(\omega + 1)^3 \right) L$$

$$< ex + \frac{2}{e} \log(n - \gamma D + 1)$$

$\square$

## A.2 Proof of Corollary 6.1

*Proof.* We have that $U_t(\lambda)$ is bounded above by $3(1 + 2B)$, its largest gradient is attained at $2/(3 + 2\eta_t/n)$ where it is bounded by $(3 + 2B)/2$. Applying Krichene et al. [28, Corollary 2] and simplifying yields the result. $\qquad\square$

## A.3 Proof of Corollary 6.2

*Proof.* $U_t(x, \lambda)$ is bounded above by $2N$ and its largest gradient is attained at $\lambda = \frac{\min\{b_t, n_t\}}{u_t(x)} \geq \frac{1}{N}$ with norm bounded by $\frac{B\exp(1/N)}{(\exp(1/N)-1)^2}$. Let $\Lambda = \{k\delta\}_{k=1}^{\lfloor 1/\delta \rfloor}$ for some $\delta \in (0, 1]$. Then we run EG on the simplex over $[N] \times \Lambda$ and with step-size $\frac{1}{2N}\sqrt{\frac{\log \frac{N}{\delta}}{2T}}$ to obtain regret compared to the best element of $[N] \times \Lambda$ of $2N\sqrt{2T\log\frac{N}{\delta}}$ [44, Theorem 2.15]. Setting $\delta = \min\left\{\frac{N(\exp(1/N)-1)^2}{B\exp(1/N)}\sqrt{\frac{2}{T}}, 1\right\}$ yields

$$\mathbb{E}\sum_{t=1}^{T} U_t(x_t, \lambda_t) \leq 2N\sqrt{2T\log(N\lfloor 1/\delta \rfloor)} + \min_{x \in [N], \lambda \in \Lambda}\sum_{t=1}^{T} U_t(x, \lambda)$$

$$\leq 2N\sqrt{2T\log\frac{N}{\delta}} + \frac{B\exp(1/N)\delta T}{(\exp(1/N)-1)^2} + \min_{x \in [N], \lambda \in (0,1]}\sum_{t=1}^{T} U_t(x, \lambda)$$

$$\leq 2N\sqrt{2T\left(\log(BT) + \max\left\{\log N, \frac{1}{N} - 2\log\left(\exp\left(\frac{1}{N}\right) - 1\right)\right\}\right)}$$

$$+ N\sqrt{2T} + \min_{x \in [N], \lambda \in (0,1]}\sum_{t=1}^{T} U_t(x, \lambda)$$

$$\leq 6N\sqrt{T\log(BNT)} + \min_{x \in [N], \lambda \in (0,1]}\sum_{t=1}^{T} U_t(x, \lambda)$$

$\qquad\square$

## A.4 Proof of Corollary 6.4

*Proof.* $U_t(x, \lambda)$ is bounded above by $e(N + B)$, its largest gradient w.r.t. $\lambda$ is attained at $\lambda = \frac{e\min\{n_t, b_t\}}{(e-1)u_t(x) + e\min\{n_t, b_t\}}$, where it is bounded by $\left(\frac{2e(N+B)}{e-1}\right)^2$, and its largest gradient w.r.t. $x$ is $e(N + B)$. Thus the function is $5e(N+B)^2$-Lipschitz w.r.t. the Euclidean norm, apart from discontinuities at $x = n_t$. Now, note that our assumption that the points $n_1, \dots, n_T$ are $\beta$-dispersed implies exactly that the functions $U_t$ are $\beta$-dispersed (c.f. Balcan et al. [9, Definition 2.1]), so the exponentially-weighted forecaster attains expected regret $\tilde{\mathcal{O}}\left(\sqrt{T\log(NT)} + (N + B)^2 T^{1-\beta}\right)$. $\qquad\square$

## A.5 Online-to-batch conversion

**Lemma A.1.** *Suppose an online learner has regret bound $R_T$ on a sequence of convex losses $\ell_{\mathbf{y}_1}, \dots, \ell_{\mathbf{y}_T} : \mathcal{X} \mapsto [0, B]$ whose data $\mathbf{y}_t$ are drawn i.i.d. from some distribution $\mathcal{D}$. If $\mathbf{x}_1, \dots, \mathbf{x}_T$ are the actions of the online learner, $\hat{\mathbf{x}} = \frac{1}{T}\sum_{t=1}^{T}\mathbf{x}_t$ is their average, and $T = \Omega\left(T_\varepsilon + \frac{B^2}{\varepsilon^2}\log\frac{1}{\delta}\right)$ for $T_\varepsilon = \min_{2R_{T'} \leq \varepsilon T'} T'$, then w.p. $\geq 1 - \delta$ we have $\mathbb{E}_{\mathbf{y}\sim\mathcal{D}}\ell_{\mathbf{y}}(\hat{\mathbf{x}}) \leq \min_{\mathbf{x}\in\mathcal{X}}\mathbb{E}_{\mathbf{y}\sim\mathcal{D}}\ell_{\mathbf{y}}(\mathbf{x}) + \varepsilon$.*

*Proof.* Apply Jensen's inequality, [12, Proposition 1], the regret bound, and Hoeffding's bound:

$$\mathbb{E}_{\mathbf{y}}\ell_{\mathbf{y}}(\hat{\mathbf{x}}) \leq \frac{1}{T}\sum_{t=1}^{T}\mathbb{E}_{\mathbf{y}}\ell_{\mathbf{y}}(\mathbf{x}_t) \leq \frac{1}{T}\sum_{t=1}^{T}\ell_{\mathbf{y}_t}(\mathbf{x}_t) + B\sqrt{\frac{2}{T}\log\frac{2}{\delta}} \leq \min_{\mathbf{x}\in\mathcal{X}}\frac{1}{T}\sum_{t=1}^{T}\ell_{\mathbf{y}_t}(\mathbf{x}) + \frac{R_T}{T} + B\sqrt{\frac{2}{T}\log\frac{2}{\delta}}$$

$$\leq \min_{\mathbf{x}\in\mathcal{X}}\mathbb{E}_{\mathbf{y}}\ell_{\mathbf{y}}(\mathbf{x}) + \frac{R_T}{T} + 2B\sqrt{\frac{2}{T}\log\frac{2}{\delta}} \tag{1}$$

$\qquad\square$

# B  b-matching

**Definition B.1.** For $\mathbf{b} \in \mathbb{R}^n_{\geq 0}$ the **b-seminorm** $\|\cdot\|_{\mathbf{b},1} : \mathbb{R}^n \mapsto \mathbb{R}_{\geq 0}$ is $\|\mathbf{x}\|_{\mathbf{b},1} = \sum_{i=1}^n \mathbf{b}_{[i]} |\mathbf{x}_{[i]}|$.

**Claim B.1.** *Given any vectors $\mathbf{x} \in \mathbb{Z}^n$ and $\mathbf{y} \in \mathbb{R}^n$, let $\tilde{\mathbf{y}} \in \mathbb{Z}^n$ be the vector whose elements are those of $\mathbf{y}$ rounded to the nearest integer. Then for all $\mathbf{b} \in \mathbb{Z}^n$ we have $\|\mathbf{x} - \tilde{\mathbf{y}}\|_{\mathbf{b},1} \leq 2\|\mathbf{x} - \mathbf{y}\|_{\mathbf{b},1}$.*

*Proof.* Let $S \subset [n]$ be the set of indices $i \in [n]$ for which $\mathbf{x}_{[i]} \geq \mathbf{y}_{[i]} \iff \tilde{\mathbf{y}}_{[i]} = \lceil \mathbf{y}_{[i]} \rceil$. For $i \in [n]\backslash S$ we have $|\mathbf{x}_{[i]} - \mathbf{y}_{[i]}| \geq 1/2 \geq |\tilde{\mathbf{y}}_{[i]} - \mathbf{y}_{[i]}|$ so it follows by the triangle inequality that

$$
\begin{aligned}
\|\mathbf{x} - \tilde{\mathbf{y}}\|_{\mathbf{b},1} &= \sum_{i \in S} \mathbf{b}_{[i]} |\mathbf{x}_{[i]} - \tilde{\mathbf{y}}_{[i]}| + \sum_{i \in [n]\backslash S} \mathbf{b}_{[i]} |\mathbf{x}_{[i]} - \tilde{\mathbf{y}}_{[i]}| \\
&\leq \sum_{i \in S} \mathbf{b}_{[i]} |\mathbf{x}_{[i]} - \mathbf{y}_{[i]}| + \sum_{i \in [n]\backslash S} \mathbf{b}_{[i]} (|\mathbf{x}_{[i]} - \mathbf{y}_{[i]}| + |\mathbf{y}_{[i]} - \tilde{\mathbf{y}}_{[i]}|) \\
&\leq \sum_{i \in S} \mathbf{b}_{[i]} |\mathbf{x}_{[i]} - \mathbf{y}_{[i]}| + 2 \sum_{i \in [n]\backslash S} \mathbf{b}_{[i]} |\mathbf{x}_{[i]} - \mathbf{y}_{[i]}| \leq 2\|\mathbf{x} - \mathbf{y}\|_{\mathbf{b},1}
\end{aligned}
$$

$\square$

**Theorem B.2.** *Suppose we have a fixed graph with $n \geq 3$ vertices and $m \geq 1$ edges.*

1. *For any cost vector $\mathbf{c} \in \mathbb{Z}^m_{\geq 0}$, any demand vector $\mathbf{b} \in \mathbb{Z}^n_{\geq 0}$, and any dual vector $\mathbf{x} \in \mathbb{R}^n$ there exists an algorithm for minimum weight perfect $\mathbf{b}$-matching that runs in time $\tilde{\mathcal{O}}(mnU(\mathbf{x}))$, where $U(\mathbf{x}) = \|\mathbf{x} - \mathbf{x}^*(\mathbf{c}, \mathbf{b})\|_{\mathbf{b},1}$ for $\mathbf{x}^*(\mathbf{c}, \mathbf{b})$ the optimal dual vector associated with $\mathbf{c}$ and $\mathbf{b}$.*

2. *There exists a poly-time algorithm s.t. for any $\delta, \varepsilon > 0$ and any distribution $\mathcal{D}$ over (cost, demand) vector pairs in $\mathbb{Z}^m_{\geq 0} \times \mathbb{Z}^n_{\geq 0}$ with respective $\ell_\infty$-norms bounded by $C$ and $B$ the algorithm takes $\mathcal{O}\left(\left(\frac{CBn}{\varepsilon}\right)^2 \log \frac{1}{\delta}\right)$ samples from $\mathcal{D}$ and returns $\hat{\mathbf{x}}$ s.t. w.p. $\geq 1 - \delta$:*

$$
\mathbb{E}_{(\mathbf{c},\mathbf{b}) \sim \mathcal{D}} \|\hat{\mathbf{x}} - \mathbf{x}^*(\mathbf{c}, \mathbf{b})\|_{\mathbf{b},1} \leq \min_{\|\mathbf{x}\|_\infty \leq C} \mathbb{E}_{(\mathbf{c},\mathbf{b}) \sim \mathcal{D}} \|\mathbf{x} - \mathbf{x}^*(\mathbf{c}, \mathbf{b})\|_{\mathbf{b},1} + \varepsilon
$$

3. *Let $(\mathbf{c}_1, \mathbf{b}_1), \ldots, (\mathbf{c}_T, \mathbf{b}_T) \in \mathbb{Z}^m_{\geq 0} \times \mathbb{Z}^n_{\geq 0}$ be an adversarial sequence of (cost, demand) vector pairs with $\ell_\infty$-norms bounded by $C$ and $B$, respectively. Then OGD with appropriate step-size has regret*

$$
\max_{\|\mathbf{x}\|_\infty \leq C} \sum_{t=1}^T \|\mathbf{x}_t - \mathbf{x}^*(\mathbf{c}_t, \mathbf{b}_t)\|_{\mathbf{b}_t,1} - \|\mathbf{x} - \mathbf{x}^*(\mathbf{c}_t, \mathbf{b}_t)\|_{\mathbf{b}_t,1} \leq CBn\sqrt{2T}
$$

*Proof.* The first result follows by Dinitz et al. [16, Theorem 31] and Claim B.1. For the third, let $\mathbf{x}_t$ be the sequence generated by running OGD [46] with step size $\frac{C}{B\sqrt{2T}}$ on the losses $U_t(\mathbf{x}) = \|\mathbf{x} - \mathbf{x}^*(\mathbf{c}_t, \mathbf{b}_t)\|_{\mathbf{b}_t,1}$ over domain $[-C, C]^n$. Since these losses are $B\sqrt{n}$-Lipschitz and the duals are $C\sqrt{n}$-bounded in Euclidean norm the regret guarantee follows from Shalev-Shwartz [44, Corollary 2.7]. For the second result, apply online-to-batch conversion to the third result, i.e. draw $T = \Omega\left(\left(\frac{CBn}{\varepsilon}\right)^2 \log \frac{1}{\delta}\right)$ samples $(\mathbf{c}_t, \mathbf{b}_t)$, run OGD as above on the resulting losses $U_t$, and set $\hat{\mathbf{x}} = \frac{1}{T}\sum_{t=1}^T \mathbf{x}_t$ to be the average of the resulting predictions $\mathbf{x}_t$. Applying Lemma A.1 yields the result. $\square$

# C Learning linear predictors with instance-feature inputs

Computational instances on which we want to run algorithms with predictions often come with instance-specific features, e.g. ones derived from text descriptions of the instance or summary statistics about related graphs or environments [27, 30]. It is thus natural to learn parameterized functions, e.g. linear mappings or neural networks, from these features to predictions. However, there has been very little work, in either the statistical or online setting, showing that such predictions are learnable. In this section we show how our framework naturally handles this setting by exploiting the convexity of compositions of convex and affine functions, resulting in the first formal guarantees for linear predictors for algorithms with predictions. While the first application to the matching problem of Dinitz et al. [16] is a straightforward extension, we also show how to handle more complicated cases, such as when the output space is constrained to probability simplices as in the page migration problem. Note we assume all feature vectors lie in the $f$-dimensional simplex; this is generally easy to accomplish by normalization. For simplicity we also only consider learning the linear transform from features to predictors and not the intercept, as the latter follows from the former by appending an extra dimension with value $1/2$ to the feature vector and doubling the bound on the norm of the linear transform.

## C.1 b-matching

Our first application for learning mappings from instance features is to the $\mathbf{b}$-matching setting. Note that the learning-theoretic results for the regular bipartite matching setting in Section 3 follow directly by setting $\mathbf{b} = \mathbf{1}_n$ for all instances, and that the learning-theoretic results of Theorems 3.1 and B.2 are also special cases of the following when $\mathbf{f} = \mathbf{1}_1$ for all instances. Note that we optimize only over $\mathbf{A} \in [-C, C]^{n \times f}$, but unlike in the $f = 1$ case the optimal $\mathbf{A}$ may be unbounded; to handle that setting, one could again use an algorithm such as KT-OCO that does not depend on knowing the set size [38].

**Theorem C.1.** *Consider the setting of Theorem B.2.*

1. *There exists a poly-time algorithm s.t. for any $\delta, \varepsilon > 0$ and any distribution $\mathcal{D}$ over (cost, demand, feature) vector triples in $\mathbb{Z}_{\geq 0}^m \times \mathbb{Z}_{\geq 0}^n \times \triangle_f$ s.t. the respective $\ell_\infty$-norms of the first two are bounded by $C$ and $B$, respectively, the algorithm takes $\mathcal{O}\left(\left(\frac{CBn}{\varepsilon}\right)^2 \left(f^2 + \log \frac{1}{\delta}\right)\right)$ samples from $\mathcal{D}$ and returns $\hat{\mathbf{A}} \in \mathbb{R}^{n \times f}$ s.t. w.p. $\geq 1 - \delta$:*

$$\mathbb{E}_{(\mathbf{c}, \mathbf{b}, \mathbf{f}) \sim \mathcal{D}} \|\hat{\mathbf{A}}\mathbf{f} - \mathbf{x}^*(\mathbf{c}, \mathbf{b})\|_{\mathbf{b}, 1} \leq \min_{\|\mathbf{A}\|_{\max} \leq C} \mathbb{E}_{(\mathbf{c}, \mathbf{b}, \mathbf{f}) \sim \mathcal{D}} \|\mathbf{A}\mathbf{x} - \mathbf{x}^*(\mathbf{c}, \mathbf{b})\|_{\mathbf{b}, 1} + \varepsilon$$

2. *Let $(\mathbf{c}_1, \mathbf{b}_1, \mathbf{f}_1), \ldots, (\mathbf{c}_T, \mathbf{b}_T, \mathbf{f}_T) \in \mathbb{Z}_{\geq 0}^m \times \mathbb{Z}_{\geq 0}^n \times \triangle_f$ be an adversarial sequence of (cost, demand, feature) vector triples s.t. the $\ell_\infty$-norms of the first two are bounded by $C$ and $B$, respectively. Then OGD with appropriate step-size has regret*

$$\max_{\|\mathbf{A}\|_{\max} \leq C} \sum_{t=1}^T \|\mathbf{A}_t \mathbf{f}_t - \mathbf{x}^*(\mathbf{c}_t, \mathbf{b}_t)\|_{\mathbf{b}_t, 1} - \|\mathbf{A}\mathbf{f}_t - \mathbf{x}^*(\mathbf{c}_t, \mathbf{b}_t)\|_{\mathbf{b}_t, 1} \leq CBnf\sqrt{2T}$$

*Proof.* For the second result let $\mathbf{A}_t$ be generated by running OGD with step-size $\frac{C}{B\sqrt{2T}}$ on the losses $U_t(\mathbf{Af}) = \|\mathbf{Af} - \mathbf{x}^*(\mathbf{c}_t, \mathbf{b}_t)\|_{\mathbf{b}_t, 1}$ over $[-C, C]^{n \times f}$. Since these are $B\sqrt{nf}$-Lipschitz and the duals are $C\sqrt{nf}$-bounded in the Euclidean norm, the regret follows from Shalev-Shwartz [44, Corollary 2.7]. For the first result, apply online-to-batch conversion to the second result, i.e. draw $T = \Omega\left(\left(\frac{CBn}{\varepsilon}\right)^2 \left(f^2 + \log \frac{1}{\delta}\right)\right)$ samples $(\mathbf{c}_t, \mathbf{b}_t, \mathbf{f}_t)$, run OGD as above on the resulting losses $U_t$, and set $\hat{\mathbf{A}} = \frac{1}{T} \sum_{t=1}^T \mathbf{A}_t$ to be the average of the resulting predictions $\mathbf{A}_t$. Applying Lemma A.1 yields the result. $\square$

## C.2 Online page migration

Using instance features for online page migration is more involved because the output space must be constrained to the product of $n$ $|\mathcal{K}|$-dimensional simplices. However, we can solve this by restricting to tensors consisting of matrices whose columns sum to one, also known as rectangular stochastic matrices. Note that the learning-theoretic results of Theorem 4.2 are special cases of the following when $\mathbf{f} = \mathbf{1}_1$ for all instances.

**Theorem C.2.** *In the setting of Theorem 4.2 let $\mathbb{S}^{n \times |\mathcal{K}| \times f}$ be the set of stacks of $|\mathcal{K}| \times f$ nonnegative matrices whose columns have unit $\ell_1$-norm.*

1. *There exists a poly-time algorithm s.t. for any $\delta, \varepsilon > 0$ and distribution $\mathcal{D}$ over request sequences $s$ of length $n$ in $\mathcal{K}$ and associated feature vectors $\mathbf{f} \in \triangle_f$ it takes $\mathcal{O}\left( \left(\frac{\gamma D}{\varepsilon}\right)^2 \left(n^2 f^2 \log |\mathcal{K}| + \log \frac{1}{\delta}\right) \right)$ samples from $\mathcal{D}$ and returns $\hat{\mathbf{A}}$ s.t. w.p. $\geq 1 - \delta$:*

$$\mathbb{E}_{(s,\mathbf{f}) \sim \mathcal{D}} U_s(\hat{\mathbf{A}}\mathbf{f}) \leq \min_{\mathbf{A} \in \mathbb{S}^{n \times |\mathcal{K}| \times f}} \mathbb{E}_{(s,\mathbf{f}) \sim \mathcal{D}} U_s(\mathbf{A}\mathbf{f}) + \varepsilon$$

2. *Let $(s_1, \mathbf{f}_1), \ldots, (s_T, \mathbf{f}_T)$ be an adversarial sequence of (request sequence, feature) pairs. Then updating the distribution $\mathbf{A}_{t[j,,k]}$ over $\triangle_{|\mathcal{K}|}$ at each (timestep,column) pair $(j,k) \in [n] \times [f]$ using EG with appropriate step-size has regret*

$$\max_{\mathbf{A} \in \mathbb{S}^{n \times |\mathcal{K}| \times f}} \sum_{t=1}^{T} U_{s_t}(\mathbf{A}_t \mathbf{f}_t) - U_{s_t}(\mathbf{A}\mathbf{f}_t) \leq \gamma D n f \sqrt{2T \log |\mathcal{K}|}$$

*Proof.* For the second result let $\mathbf{A}_t$ be the sequence generated by running $nf$ EG algorithms with step size $\sqrt{\frac{\log |\mathcal{K}|}{2\gamma^2 D^2 T}}$ on the losses $U_{s_t}(\mathbf{A}\mathbf{f})$ over $\mathbb{S}^{n \times |\mathcal{K}| \times f}$. Since these losses are $\gamma D$-Lipschitz and the maximum entropy over the simplex is $\log |\mathcal{K}|$, the regret guarantee follows from [44, Theorem 2.15]. For the first result, apply standard online-to-batch conversion to the second result, i.e. draw $T = \Omega\left( \left(\frac{\gamma D}{\varepsilon}\right)^2 \left(n^2 f^2 \log |\mathcal{K}| + \log \frac{1}{\delta}\right) \right)$ samples $(\mathbf{s}_t, \mathbf{f}_t)$, run EG on the resulting losses $U_{s_t}(\mathbf{A}\mathbf{f}_t)$ as above, and set $\hat{\mathbf{A}} = \frac{1}{T} \sum_{t=1}^{T} \mathbf{A}_t$ to be the average of the resulting actions $\mathbf{A}_t$. Applying Lemma A.1 yields the result. $\qquad\square$

We can further also show a result in the perhaps more-natural setting where the linear predictor $\mathbf{A}$ is the same for each element in the sequence, and maps directly from features to the $|\mathcal{K}|$-simplex. Notably, the linear auto-regressive setting, in which we want a linear map from the past $k$ sequence elements to a probabilistic prediction of the next one, is covered by this result if we allow the features to be $k|\mathcal{K}|$-dimensional concatenations of $k$ one-hot $|\mathcal{K}|$-length vectors.

**Theorem C.3.** *In the setting of Theorem 4.2 let $\mathbb{S}^{a \times b}$ be the set of $a \times b$ nonnegative matrices whose columns have unit $\ell_1$-norm.*

1. *There exists a poly-time algorithm s.t. for any $\delta, \varepsilon > 0$ and distribution $\mathcal{D}$ over request sequences $s$ of length $n$ in $\mathcal{K}$ and associated feature sequence $\mathbf{F}^T \in \mathbb{S}^{f \times n}$ it takes $\mathcal{O}\left( \left(\frac{\gamma D}{\varepsilon}\right)^2 \left(n^2 f^2 \log |\mathcal{K}| + \log \frac{1}{\delta}\right) \right)$ samples from $\mathcal{D}$ and returns $\hat{\mathbf{A}}$ s.t. w.p. $\geq 1 - \delta$:*

$$\mathbb{E}_{(s,\mathbf{F}) \sim \mathcal{D}} U_s(\mathbf{F}\hat{\mathbf{A}}^T) \leq \min_{\mathbf{A} \in \mathbb{S}^{|\mathcal{K}| \times f}} \mathbb{E}_{(s,\mathbf{F}) \sim \mathcal{D}} U_s(\mathbf{F}\mathbf{A}^T) + \varepsilon$$

2. *Let $(s_1, \mathbf{F}_1), \ldots, (s_T, \mathbf{F}_T)$ be an adversarial sequence of (request sequence, feature sequence) pairs. Then updating the distribution $\mathbf{A}_{t[,k]}$ over $\triangle_{|\mathcal{K}|}$ at each column $k \in [f]$ has regret*

$$\max_{\mathbf{A} \in \mathbb{S}^{|\mathcal{K}| \times f}} \sum_{t=1}^{T} U_{s_t}(\mathbf{F}_t \mathbf{A}_t^T) - U_{s_t}(\mathbf{F}_t \mathbf{A}^T) \leq \gamma D f \sqrt{2T \log |\mathcal{K}|}$$

*Proof.* For the second result let $\mathbf{A}_t$ be the sequence generated by running $f$ EG algorithms with step size $\sqrt{\frac{\log |\mathcal{K}|}{2\gamma^2 D^2 T}}$ on the losses $U_{s_t}(\mathbf{F}\mathbf{A}^T)$ over $\mathbb{S}^{|\mathcal{K}| \times f}$. Since these losses are $\gamma D$-Lipschitz and the maximum entropy over the simplex is $\log |\mathcal{K}|$, the regret guarantee follows from [44, Theorem 2.15]. For the first result, apply standard online-to-batch conversion to the second result, i.e. draw $T = \Omega\left( \left(\frac{\gamma D}{\varepsilon}\right)^2 \left(f^2 \log |\mathcal{K}| + \log \frac{1}{\delta}\right) \right)$ samples $(\mathbf{s}_t, \mathbf{f}_t)$, run EG on the resulting losses $U_{s_t}(\mathbf{F}_t \mathbf{A}^T)$ as above, and set $\hat{\mathbf{A}} = \frac{1}{T} \sum_{t=1}^{T} \mathbf{A}_t$ to be the average of the resulting actions $\mathbf{A}_t$. Applying Lemma A.1 yields the result. $\qquad\square$

## D   Faster graph algorithms with predictions

In this section we compare to the results of Chen et al. [13], who analyze several prediction-based graph algorithms, including one with an improved prediction-dependent runtime for the matching approach of Dinitz et al. [16] and a prediction-dependent bound for single-source shortest path. From the learnability perspective, they observe two important error metrics in the analysis of graph algorithms with predictions: the $\ell_1$-metric of Dinitz et al. [16] measuring the $\ell_1$-norm between the prediction and a ground truth vector such as the dual and the $\ell_\infty$-metric measuring the $\ell_\infty$-norm between the same quantities. In the first case their setting and results are equivalent to those of Dinitz et al. [16], so we improve upon this by a factor of $\mathcal{O}(d)$, where $d$ is the dimension of the hint.

To analyze the $\ell_\infty$ case, we start by showing that—as in the $\ell_1$ case—we can round integer vectors with only a multiplicative factor loss:

**Claim D.1.** *Given any vectors $\mathbf{x} \in \mathbb{Z}^n$ and $\mathbf{y} \in \mathbb{R}^n$, let $\tilde{\mathbf{y}} \in \mathbb{Z}^n$ be the vector whose elements are those of $\mathbf{y}$ rounded to the nearest integer. Then we have $\|\mathbf{x} - \tilde{\mathbf{y}}\|_\infty \le 2\|\mathbf{x} - \mathbf{y}\|_\infty$.*

*Proof.* Let $S \subset [n]$ be the set of indices $i \in [n]$ for which $\mathbf{x}_{[i]} \ge \mathbf{y}_{[i]} \iff \tilde{\mathbf{y}}_{[i]} = \lceil \mathbf{y}_{[i]} \rceil$. For $i \in [n] \backslash S$ we have $|\mathbf{x}_{[i]} - \mathbf{y}_{[i]}| \ge 1/2 \ge |\tilde{\mathbf{y}}_{[i]} - \mathbf{y}_{[i]}|$ so it follows by the triangle inequality that

$$
\begin{aligned}
\|\mathbf{x} - \tilde{\mathbf{y}}\|_\infty &= \max\left\{ \max_{i \in S} |\mathbf{x}_{[i]} - \tilde{\mathbf{y}}_{[i]}|, \max_{i \in [n]\backslash S} |\mathbf{x}_{[i]} - \tilde{\mathbf{y}}_{[i]}| \right\} \\
&\le \max\left\{ \max_{i \in S} |\mathbf{x}_{[i]} - \mathbf{y}_{[i]}|, \max_{i \in [n]\backslash S} |\mathbf{x}_{[i]} - \mathbf{y}_{[i]}| + |\mathbf{y}_{[i]} - \tilde{\mathbf{y}}_{[i]}| \right\} \\
&\le \max\left\{ \max_{i \in S} |\mathbf{x}_{[i]} - \mathbf{y}_{[i]}|, 2 \max_{i \in [n]\backslash S} |\mathbf{x}_{[i]} - \mathbf{y}_{[i]}| \right\} \le 2\|\mathbf{x} - \mathbf{y}\|_\infty
\end{aligned}
$$

$\square$

We are thus able to also use online convex optimization in this setting and apply the rounded outputs to graph algorithms. In particular, we can use regular OGD to improve upon the $\ell_\infty$-learnability result of Chen et al. [13] by a factor of $\mathcal{O}(d^2)$, where $d$ is the dimension of the prediction:

**Theorem D.1.** *Consider any graph algorithm with optimal $d$-dimensional $M$-bounded predictions $\mathbf{h}(c)$ associated with every instance $c$.*

*1. There exists a poly-time algorithm s.t. for any $\delta, \varepsilon > 0$ and distribution $\mathcal{D}$ over instances it takes $\mathcal{O}\left( \left(\frac{M}{\varepsilon}\right)^2 \left( d + \log \frac{1}{\delta} \right) \right)$ samples from $\mathcal{D}$ and returns $\hat{\mathbf{h}} \in \mathbb{R}^d$ s.t. w.p. $\ge 1 - \delta$*

$$
\mathbb{E}_{c \sim \mathcal{D}} \|\hat{\mathbf{h}} - \mathbf{h}(c)\|_\infty \le \min_{\|\mathbf{h}\|_\infty \le M} \mathbb{E}_{c \sim \mathcal{D}} \|\mathbf{h} - \mathbf{h}(c)\|_\infty + \varepsilon
$$

*2. Let $c_1, \dots, c_T$ be an adversarial sequence of instances. Then OGD with appropriate step-size achieves regret*

$$
\max_{\|\mathbf{h}\|_\infty \le M} \sum_{t=1}^{T} \|\mathbf{h}_t - \mathbf{h}(c_t)\|_\infty - \|\mathbf{h} - \mathbf{h}(c_t)\|_\infty \le M\sqrt{2dT}
$$

*Proof.* The proof is the same as for the last two results of Theorem 5.1 in the special case $f = 1$. $\square$

# E   Permutation predictions for non-clairvoyant scheduling

Finally, we discuss the the applicability of our framework to the results in Lindermayr and Megow [33], who study how to prioritize among $n$ jobs by predicting the best permutation of them under weights $\mathbf{w} \in \mathbb{R}_{\geq 0}^n$ and processing requirements $\mathbf{p} \in \mathbb{R}_{\geq 0}^n$ that are only known after completion. Ignoring robustness-consistency tradeoffs and terms that do not depend on the prediction, they show that in several settings the competitive ratio depends linearly on the following error of an $n \times n$ permutation matrix $\mathbf{X}$:

$$U_{\mathbf{w},\mathbf{p}}(\mathbf{X}) = \mathrm{Tr}(\mathbf{X}\mathbf{w}((\mathbf{U} \odot \mathbf{X})\mathbf{p})^T) = \mathrm{Tr}((\mathbf{U} \odot \mathbf{X})^T \mathbf{X}\mathbf{w}\mathbf{p}^T)$$

where $\mathbf{U} \in \{0,1\}^{n \times n}$ is upper triangular. The above expression is derived from the third equation in the proof of Theorem 4.1 of Lindermayr and Megow [33] for the case of $z = 1$ sample; we construct the matrix form to reason about its online learnability.

Naively, a sequence of bounded functions of permutations is *computationally inefficiently* learnable by using randomized EG over the $n!$ experts corresponding to each permutation:

**Theorem E.1.** *Consider the setting of Lindermayr and Megow [33] with $n$ jobs with $W$-bounded weights and $P$-bounded processing times. Let $\mathbb{P}^{n \times n}$ be the set of $n \times n$ permutation matrices.*

1. *There exists an algorithm that s.t. for any $\delta, \varepsilon > 0$ and distribution $\mathcal{D}$ over weights and processing requirements it takes $\mathcal{O}\left(\left(\frac{WPn}{\varepsilon}\right)^2 \left(n \log n + \log \frac{1}{\delta}\right)\right)$ samples from $\mathcal{D}$ and returns a discrete distribution $\hat{\mathbf{x}} \in \triangle_{n!}$ over $\mathbb{P}^{n \times n}$ such that*

$$\mathbb{E}_{\mathbf{X} \sim \hat{\mathbf{x}}} \mathbb{E}_{(\mathbf{w},\mathbf{p}) \sim \mathcal{D}} U_{\mathbf{w},\mathbf{p}}(\mathbf{X}) \leq \min_{\mathbf{X} \in \mathbb{P}^{n \times n}} \mathbb{E}_{(\mathbf{w},\mathbf{p}) \sim \mathcal{D}} U_{\mathbf{w},\mathbf{p}}(\mathbf{X}) + \varepsilon$$

2. *Let $(\mathbf{w}_1, \mathbf{p}_1), \ldots, (\mathbf{w}_T, \mathbf{p}_T)$ be an adversarial sequence of job (weight, processing requirement) pairs. Then running EG with appropriate step-size over $\triangle_{|\mathbb{P}^{n \times n}|}$ has regret*

$$\mathbb{E} \max_{\mathbf{X} \in \mathbb{P}^{n \times n}} \sum_{t=1}^{T} U_{\mathbf{w}_t, \mathbf{p}_t}(\mathbf{X}_t) - U_{\mathbf{w}_t, \mathbf{p}_t}(\mathbf{X}) \leq WPn\sqrt{2nT \log n}$$

*where the expectation is over the randomness of the algorithm.*

*Proof.* For the second result let $\mathbf{x}_t \in \triangle_{n!}$ be the sequence generated by running EG with step-size $\sqrt{\frac{\log(n!)}{2T}}$ over the $n!$ experts corresponding to each element of $\mathbb{P}^{n \times n}$. Then the sequence of permutation matrices $\mathbf{X}_t \sim \mathbf{x}_t$ sampled from this distribution satisfies the guarantee on the expected regret [44, Corollary 2.14] since $\log(n!) \leq n \log n$ and $U_{\mathbf{w},\mathbf{p}}$ is $WPn$-bounded. The first result follows by applying online-to-batch conversion to this sequence, i.e. we draw $T = \Omega\left(\left(\frac{WPn}{\varepsilon}\right)^2 \left(n \log n + \log \frac{1}{\delta}\right)\right)$ samples $(\mathbf{w}_t, \mathbf{p}_t)$, run randomized EG as above, and set $\hat{\mathbf{x}} = \frac{1}{T} \sum_{t=1}^{T} \mathbf{x}_t$ to be the average of the resulting distributions $\mathbf{x}_t$. Applying Lemma A.1 yields the result.  □

The sample complexity guarantee resulting from online-to-batch conversion matches that of Lindermayr and Megow [33], except that the output is a distribution over permutation matrices so the error is in expectation over that distribution. However, randomized EG is incredibly inefficient due to the need to store and sample from a distribution over $n!$ variables. Another way of learning over permutation matrices is to run an online learning algorithm over the set of doubly stochastic matrices [21]. When the losses are linear functions of the permutation matrices this is yields efficient low-regret algorithms because each doubly stochastic matrix corresponds to a *small* convex combination of permutation matrices, i.e. a distribution from which one can sample an action. However, the losses $U_{\mathbf{w},\mathbf{p}}$ are nonlinear and so a different approach is needed.