# OpenReview forum: "Learning Predictions for Algorithms with Predictions"
_NeurIPS.cc/2022/Conference — NeurIPS 2022 Accept_

### Official Review · Reviewer_zeSS · 2022-07-05

**Rating:** 7
**Confidence:** 5
**Soundness:** 4 excellent
**Presentation:** 4 excellent
**Contribution:** 3 good

**Summary:**

The paper studies learning-augmented algorithms, mostly focused on the online setting. A recent line of works establishes that one can improve upon classic, worst-case guarantees if the algorithm is furnished with good predictions. However, the question of when and how such good predictions can be learned from data (efficiently) is still open.

This work presents a fairly general approach to answering this question. Prior work shows that the performance of a learning-based algorithm can be bounded by some error measure of the prediction (compared with the best prediction). This paper shows that if there's an efficiently-optimizable upper bound of the error measure, then techniques from online learning (in particular, online convex optimization) can be applied to learn good predictions.

It is shown that this general learning-theoretic framework can be applied to matching, paging migration, scheduling and ski-rental.

**Questions:**

On the technical side, can the dimension-dependence (on n) of theorem 3.1 be improved? I think using OGD on the L-infty ball constraint may be loose. Note that one sqrt n factor comes from Line 181, where the L-infty constraint is converted to L2, allowing one to apply the OGD guarantee. The author(s) may want to look into the relevant literature on online mirror descent; e.g., “On the Universality of Online Mirror Descent” (https://arxiv.org/abs/1107.4080)


**Limitations:**

In general, I feel that this approach has more potential, to be applied to more problems. Here are two concrete suggestions:

* “Permutation Predictions for Non-Clairvoyant Scheduling” (SPAA 22) studies a new prediction model for non-clairvoyant scheduling, which provides a relative order of jobs. It seems that the continuous relaxation technique of section 4 can be used to learn this permutation. I believe this could be more broadly interesting, to other online problems, such as paging and   graph problems (where vertices/edges arrive online in some order, and it would help if you can learn this order).

* “Faster Fundamental Graph Algorithms via Learned Predictions” (ICML 22) considers speeding up other (offline) graph problems, not just matching. The techniques from section 3 can probably be used in those settings.

Finally, I'd like to see whether the regret bounds from section 3 can be improved.

**Strengths And Weaknesses:**

Strengths
----
The paper is solid on a technical level and the theoretical claims are sound.

Overall I find the paper very well written.

To the best of my knowledge, the results are novel. Given the recent spate of work on algorithms with predictions, I believe it has potential to make further impact

Weaknesses
---
I am not convinced that the paper exhausts all its potential. Particularly, there should be some more problems where this general framework can be applied. I suggest a few in a later section. I encourage the author(s) to consider them in the final version or in a followup work.


In comparison with some of the prior work, the paper  does not provide experimental evaluations.

Minor and missing references
---
Related work: “Faster Fundamental Graph Algorithms via Learned Predictions” (ICML 22) also studies bipartite matching, extending Dinitz et al.

Theorem 3.1. I suggest the author(s) switch the order of the third and second result, since the second result is obtained from the third (using online-to-batch conversion), so it’s natural to present the third first.

---

> ### Author Response · Authors · 2022-08-02
> **Response to Reviewer zeSS**
>
> Thank you for your positive review and insightful suggestions. We hope to address your comments and questions below:
> 1. [*I am not convinced that the paper exhausts all its potential. Particularly, there should be some more problems where this general framework can be applied.*]
> In the appendix of the revision we have added discussion of how our framework applies to several additional settings—learning linear mappings from instance features, online scheduling, and the two you suggested below—and will include parts of these in the main text if accepted.
> 2. [*In comparison with some of the prior work, the paper does not provide experimental evaluations.*]
> Given the breadth of our potential learning-theoretic applications, our focus was on demonstrating how to apply our framework to proving guarantees for different problems.
> 3. [*Theorem 3.1. I suggest the author(s) switch the order of the third and second result*]
> We considered doing this, for the reasons you mention, but also wanted to match the cadence in past work (e.g. Dinitz et al. (2021)) of the runtime guarantee followed by sample complexity. We will think about this further.
> 4. [*can the dimension-dependence (on n) of theorem 3.1 be improved?*]
> Thank you for sharing this insight. In-fact yes, a factor Õ(\sqrt{n}) can be removed from the regret by using the p-norm algorithm rather than online gradient descent, which also leads to an Õ(n) improvement in the sample complexity. Note that the new linear dependence on n is likely optimal according to the table in the reference you pointed to (in our case p_1=p_2=\infty).
> 5. [*“Permutation Predictions for Non-Clairvoyant Scheduling” (SPAA 22)*]
> Thank you for this suggestion; we have noted the paper in the introduction and added a larger discussion in the appendix. We believe the error metric in this paper can be written as Trace((UoX)^TXwp^T), where X is the permutation prediction, U is the upper triangular matrix of ones, w is the vector of job weights, and p is the vector of processing requirements. Naively learning permutations can be treated as an experts problem over n! experts, which yields sublinear regret and can be converted into a sample complexity bound that matches the paper’s, but the algorithm is intractable. If the objective were linear in X we could learn over the set of stochastic matrices, as in Helmbold & Warmuth (2009), but in this case the objective is more complicated and so more work may be required.
> 6. [*“Faster Fundamental Graph Algorithms via Learned Predictions” (ICML 22)*]
> Thank you for this suggestion; we have noted the paper in the introduction and added a larger discussion in the appendix. In this case one of the error metrics is the same as the one for the learned duals problem, and so our framework of course applies and yields the same improvement. We have added a discussion of how guarantees targeting their infinity-norm metric can also be obtained using our approach, obtaining O(d^2) improvement over their sample complexity result, where d is the dimension of the prediction. However, many of the objectives in their paper contain the product of the 1-norm and infinity-norm metrics, which it does not show learning guarantees for. Since the product is non-convex, we are also not sure whether our results extend to that setting.
>
> &nbsp;
> **References:**
> 1. Dinitz, Im, Lavastida, Moseley, Vassilvitskii. *Faster matchings via learned duals*. NeurIPS 2021.
> 2. Helmbold, Warmuth. *Learning permutations with exponential weights*. JMLR 2009.

---

> > ### Comment · Reviewer_zeSS · 2022-08-05
> > **Re**
> >
> > Thank you for the response! I believe my questions and concerns have been addressed. I would recommend accept.

---

### Official Review · Reviewer_58f3 · 2022-07-10

**Rating:** 5
**Confidence:** 4
**Soundness:** 2 fair
**Presentation:** 2 fair
**Contribution:** 3 good

**Summary:**

The paper studies the learnability of predictions in the field of algorithms with predictions. The authors propose a general framework to learn predictions, and show that the framework can be applied to bipartite matching and page migration. In each setting, the proposed learning method obtains statistical guarantees. Furthermore, they show that the learning framework can help to tune the robustness-consistency tradeoffs in some problems, e.g. scheduling and ski rental.

**Questions:**

- The ``OnlineAlgorithm" in Algorithm 1 is unclear to me. Does this refer to the online gradient descent algorithm?

- Figure 1 is too far from where it is referenced.

- The submission should not have the appendices.

**Limitations:**

I didn't see any potential negative societal impact.

**Strengths And Weaknesses:**

Strength:
- The learnability of predictions is a piece in the learning-augmented model which receives little attention but is very necessary. The paper makes contributions in this part. It not only improves the sample complexity given in the previous work when the instance (of bipartite matching) is sampled from a fixed distribution, but also presents theoretical bounds of the sample complexity and the average regret when instances are from an adversarial sequence. To the best of my knowledge, this is the first regret bound in the learning-augmented framework.

Weakness:
- The paper has theoretical contributions but is not that technically deep. It builds on several existing theorems to give the statistical guarantees, which makes the results technically simple.

- There is no experiment. In (Dinitz et al, NeurIPS 2021), experiments were provided to investigate the empirical performance. Since the paper introduces a new learning setting, it is expected that there are experiments to show that the new learning framework can perform better than the previous learning algorithm (empirical risk minimization) on some data sets.

---

> ### Author Response · Authors · 2022-08-02
> **Response to Reviewer 58f3**
>
> Thank you for your positive review. We hope to address your comments and questions below.
> 1. [*The paper has theoretical contributions but is not that technically deep.*]
> The many results that follow from the general framework after it has been constructed is a testament to its potential: we view as a positive aspect the fact that in future algorithms with predictions work, researchers will be able to apply it directly to new computational guarantees to obtain learning-theoretic results. Note also that several components of the work, e.g. the derivation of a learnable upper bound for online page migration, involved significant novel analysis.
> 2. [*There is no experiment.*]
> Given the breadth of our potential learning-theoretic applications, our focus was on demonstrating how to apply our framework to proving guarantees for different problems.
> 3. [*The ``OnlineAlgorithm" in Algorithm 1 is unclear to me. Does this refer to the online gradient descent algorithm?*]
> It refers to the application-specific online algorithm optimizing the upper bounds on the cost of the application’s AlgorithmWithPrediction. The full list is in the second-to-last column of Table 1. So for example in the bipartite matching application it was indeed online gradient descent (prior to the update to p-norm following the suggestion by Reviewer zeSS), but in page migration we use exponentiated gradient.
> 4. [*Figure 1 is too far from where it is referenced.*]
> This was done in the submission to save space, but we should have more room to put it closer if the paper is accepted.
> 5. [*The submission should not have the appendices.*]
> This is allowed this year by the [NeurIPS 2022 FAQ](https://neurips.cc/Conferences/2022/PaperInformation/NeurIPS-FAQ): “You can include appendices with the main submission file, or you can include them as a separate file in the supplementary materials.”

---

### Official Review · Reviewer_wD8D · 2022-07-11

**Rating:** 6
**Confidence:** 2
**Soundness:** 3 good
**Presentation:** 4 excellent
**Contribution:** 3 good

**Summary:**

This paper proposes a general meta-framework for improving algorithms with predictions (e.g. an initial guess). The authors propose a 2 step procedure to iteratively learn better predictions: 1. upper bound the problem's cost (or tradeoff parameter) using the prediction (e.g. distance between prediction and global optimum), then 2. update the prediction using this cost estimate via standard online learning, with the goal of minimizing expected regret and sample complexity. This framework is used to derive improved guarantees across a wide range of algorithms such as matching and scheduling.


**Questions:**

- How tight are the regret/sample complexity bounds in practice on real problem instances?

**Limitations:**

This paper describes limitations as opportunities for future work, but does not address any potential negative social impact.

**Strengths And Weaknesses:**

## Originality
- This paper presents several novel analyses under a unified framework, and cites previous work where appropriate

## Quality/Significance
- The proposed framework is interesting, well-motivated, and has potential for many follow up works
- In the beginning of Section 4, the authors claim an upper bound on parameter $q$. However the assumption in [17] implies a lower bound on $q$. I'm unsure how much this possible bug affects the rest of the analysis.
- This paper has clear, elegant theoretical contributions. However, it would be stronger if it also contained experiments that verify the regret guarantees, sample complexity bounds, and general efficacy of the new algorithms

## Clarity
- Well organized and a pleasure to read
- Clear statements of results and intuitions behind them
- Typo: on Line 89 ", nut they also" should be ", but they also"

[17] Piotr Indyk, Frederik Mallmann-Trenn, Slobodan Mitrovic, and Ronitt Rubinfeld. Online page migration with ML advice. In Proceedings of the 25th International Conference on Artificial Intelligence and Statistics, 2022.

---

> ### Author Response · Authors · 2022-08-02
> **Response to Reviewer wD8D**
>
> Thank you for your positive review. We hope to address your comments and questions below:
> 1. [*the authors claim an upper bound on parameter q. However the assumption in [17] implies a lower bound on q.*]
> In [17, Assumption 1], q is defined as the smallest number lower-bounded by the proportion of mismatches in any interval of length \epsilon\*D. Thus it can be equivalently defined as the maximum over all intervals of length \epsilon\*D of the proportion of mismatches in that interval, which is the definition we use (c.f. the equation below the “Deriving an upper bound” paragraph). We want to upper-bound this quantity since (1) it bounds the maximum number of mistakes made in any interval and (2) an upper bound on q implies an upper bound on the competitive ratio [17, Theorem 1], which we want to be small.
> 2. [*it would be stronger if it also contained experiments that verify the regret guarantees, sample complexity bounds, and general efficacy of the new algorithms*]
> Given the breadth of our potential learning-theoretic applications, our focus was on demonstrating how to apply our framework to proving guarantees for different problems.
> 3. [*How tight are the regret/sample complexity bounds in practice on real problem instances?*]
> Sample complexity guarantees are often not tight in practice, e.g. in Dinitz et al. (2021) the number of samples need to learn a good prediction for bipartite matching was much smaller than their bounds. However, they are still useful for understanding what is and is not learnable as well as broad dependencies on problem parameters.
>
> &nbsp;
> **References:**
> 1. Dinitz, Im, Lavastida, Moseley, Vassilvitskii. *Faster matchings via learned duals*. NeurIPS 2021.

---

### Official Review · Reviewer_br4H · 2022-07-13

**Rating:** 7
**Confidence:** 4
**Soundness:** 3 good
**Presentation:** 3 good
**Contribution:** 3 good

**Summary:**

This paper gives a framework for learning parameters in learning-augmented algorithms. First, an upper bound is computed on the objective of interest such that this upper bound function can be optimized. If the upper bound is reasonably tight, this is also a reasonable optimization for the actual cost. Then, the parameters optimizing the upper bound of the cost function are used as learned parameters in the learning-augmented algorithm.

Two forms of this optimization are considered in this paper: sample complexity bounds for inputs drawn from a distribution and regret bounds from online learning for adversarial inputs. In terms of applications, this idea is applied to obtain new sample complexity and online learning bounds for several problems: speeding up matching algorithms using a warm start by learned duals, online page migration, obtaining the optimal consistency-robustness tradeoff in problems like ski rental and scheduling, etc.

**Questions:**

I do not have specific questions. This is a nice paper overall and should be accepted.

**Limitations:**

There is some discussion about limitations and future work at the end of the paper. It reads somewhat generic though, mostly focusing on some broad future directions for the entire field of learning-augmented algorithms rather than being specific to this paper.

**Strengths And Weaknesses:**

Strengths:

1. I like the general area of learning-augmented algorithms and data driven algorithms, and this paper makes an important contribution to this area. There is relatively little work on the learnability of predicted parameters. Instead, the focus has been on developing (robust) algorithms to exploit (noisy) predictions generated by a black box learner. This is reasonable for certain situations, e.g., if one is using a deep neural network for the purpose of generating parameters. But, in situations where we do have well-understood theoretical abstractions such as PAC and online learning, it makes sense to think of a white box approach as proposed in this paper.

2. The framework proposed is quite general and intuitive and can be applied broadly to multiple learning paradigms. In particular, the use of online learning in making predictions about algorithmic parameters is novel (to the best of my knowledge).

Weaknesses:

1. One might complain that there is no killer application here. In particular, the improvements in sample complexity bounds are relatively modest.

---

> ### Author Response · Authors · 2022-08-02
> **Response to Reviewer br4H**
>
> Thank you for your positive review. We believe the killer application of our work is in providing learning guarantees for the broad field of algorithms with predictions, for which we have provided more evidence in the revision. Note also that, following a suggestion by another reviewer, our sample complexity improvement over past work for the bipartite matching problem is now O(n) rather than O(\log{n}), where n is the number of vertices. We can also show O(d^2) improvement over Theorem 6.2 of Chen et al. (2022).
>
> &nbsp;
> **References:**
> 1. Chen, Silwal, Vakilian, Zhang. *Faster fundamental graph algorithms via learned predictions*. ICML 2022.

---

### Author Response · Authors · 2022-08-02
**General response**

Thank you to the reviewers for the positive reviews and many helpful suggestions. Our submission studies the learning component of algorithms with predictions, a very popular and impactful area in theory and practice, and proposes a general framework for proving learning-theoretic guarantees for these problems. We thus believe the work has potential for great future impact in providing end-to-end guarantees—ones encompassing both computation and learning—for this growing field. In further support of this, and following some suggestions from the reviewers, we have uploaded a new draft containing discussions of additional algorithms with predictions that our work can provide learning guarantees for, as well as other revisions and corrections. If accepted, we hope to use the additional space to include some of these new applications in the main text, which we believe strengthens the paper.

&nbsp;
**Summary of the main changes:**
1. In Section 3, following a suggestion by Reviewer zeSS, we updated the learning algorithm used in order to remove a Õ(\sqrt{n}) factor from the regret for learning duals for bipartite matching, where n is the number of vertices in the graph. This also reduces the sample complexity by a factor of Õ(n), so we now improve upon the past work by a factor of O(n).
2. In Section 4, we corrected the regret to be O(n*\sqrt{T}) rather than O(\sqrt{n*T}). We apologize for this error.
3. In Section C.1 we consider the case where computational instances come with feature vectors and we wish to learn linear mappings from them to predictions, a setting which generalizes our existing results for fixed predictions. Here we extend our results for bipartite matching and page migration and also show a new application to the online scheduling work of Lattanzi et al. (2020). Learning from instance features is an important setting for algorithms with predictions and we believe this is the first work that shows learning guarantees for it.
4. In Section C.2 we consider the paper by Chen et al. (2022) suggested by Reviewer zeSS and show that for one of their settings our framework provides the first online algorithms and obtains an O(d^2) improvement in sample complexity.
5. In Section C.3 we consider the paper by Lindermayr & Megow (2022) suggested by Reviewer zeSS and show that our framework can also match their learning-theoretic guarantees, albeit with an inefficient algorithm.

&nbsp;
**References:**
1. Chen, Silwal, Vakilian, Zhang. *Faster fundamental graph algorithms via learned predictions*. ICML 2022.
2. Lattanzi, Lavastida, Moseley, Vassilvitskii. *Online scheduling via learned weights*. SODA 2020.
3. Lindermayer, Megow. *Permutation predictions for non-clairvoyant scheduling*. SPAA 2022.

---

### Meta-Review · Area_Chair_7vUC · 2022-08-21

**Recommendation:** Accept
**Confidence:** Certain

**Metareview:**

The paper introduces a general design approach for algorithms that learn predictors. This is achieved by identifying a functional dependence of the performance measure on the prediction quality, and applying techniques from online learning to learn predictors against adversarial instances, tune robustness-consistency trade-offs, and obtain new statistical guarantees. The problem is well-motivated and the proposed solution is general and interesting. Majority of the reviewers' concerns are addressed by the rebuttal. The paper will be significantly stronger if the authors benefit from the additional page to incorporate the feedback into the main paper.

**Award:**

No

---

### Decision · Program_Chairs · 2022-09-14

Accept